# On the difference between thermalization in open and isolated quantum systems: a case study

Archak Purkayastha[1,*], Giacomo Guarnieri[2,3], Janet Anders[4,5], Marco Merkli[6]

[1] Department of Physics, Indian Institute of Technology, Hyderabad 502284, India.
[2] Department of Physics and INFN - Sezione di Pavia, University of Pavia, Via Bassi 6, 27100, Pavia, Italy.
[3] Dahlem Center for Complex Quantum Systems, Freie Universität Berlin, 14195 Berlin, Germany.
[4] Physics and Astronomy, University of Exeter, Exeter EX4 4QL, United Kingdom.
[5] Institut für Physik, Potsdam University, 14476 Potsdam, Germany.
[6] Department of Mathematics and Statistics, Memorial University of Newfoundland, St. John's, NL, Canada A1C 5S7
* archak.p@phy.iith.ac.in

September 19, 2024

## Abstract

Thermalization of isolated and open quantum systems has been studied extensively. However, being the subject of investigation by different scientific communities and being analysed using different mathematical tools, the connection between the isolated (IQS) and open (OQS) approaches to thermalization has remained opaque. Here we demonstrate that the fundamental difference between the two paradigms is the order in which the long time and the thermodynamic limits are taken. This difference implies that they describe physics on widely different time and length scales. Our analysis is carried out numerically for the case of a double quantum dot (DQD) coupled to a fermionic lead. We show how both OQS and IQS thermalization can be explored in this model on equal footing, allowing a fair comparison between the two. We find that while the quadratically coupled (free) DQD experiences no isolated thermalization, it of course does experience open thermalization. For the non-linearly interacting DQD coupled to fermionic lead, we show by characterizing its spectral form factor and level spacing distribution, that the system falls in the twilight zone between integrable and non-integrable regimes, which we call partially non-integrable. We further evidence that, despite being only partially non-integrable and thereby falling outside the remit of the standard eigenstate thermalization hypothesis, it nevertheless experiences IQS as well as OQS thermalization.

# 1 Introduction

Common experience shows that a system kept in contact with surroundings at a given temperature eventually thermalizes to that temperature. This forms the basis of much of standard thermodynamics and statistical physics. Yet, how such a process can consistently arise from the principles of quantum mechanics has been one of the fundamental questions, with a long history across physics [1–13] and mathematics [14–22]. Two classes of approaches to this problem have now become well-established. The first is the open quantum system (OQS) approach. In this approach, the surrounding environment (bath) of the system is usually modelled via a continuum of bosonic or fermionic modes, initially in a Gibbs state with a given temperature, uncorrelated with the system. Equations describing the dynamics of the system are then derived by formally integrating out the bath degrees of freedom. This can be done in any of the standard approaches to OQS, like quantum master equations [23, 24], non-equilibrium Green's functions, Feynman-Vernon influence functional, Keldysh-Schwinger path integrals [25–27] and quantum Langevin equations [12, 13, 28–32]. The OQS approach then aims to show that, irrespective of the initial state of the system, in the long-time limit, the system approaches a steady state consistent with the initial temperature of the bath. This state of the system is expected to be the marginal of the global Gibbs state of the system and the bath [7–13]. This is the starting point of many strong-coupling approaches to quantum thermodynamics, where such a state is called the mean-force Gibbs state (MGS) [33–35]. To the leading order for a small system-bath coupling, the MGS is the Gibbs state of the system with the temperature of the bath, which is completely consistent with the notion of thermalization in standard statistical physics. Considerable efforts have historically been dedicated to derive physically consistent weak system-bath coupling quantum master equations for the system degrees of freedom, satisfying convergence to the Gibbs state of the system [23]. Only recently, the fundamental limitations of such an approach have been appreciated [36,37], and an improved quantum master equation considering convergence to MGS has been derived [38].

In a second well-established approach to describing the process of thermalization, no demarcation is made between the system and the surroundings. The whole set-up is taken to be isolated and evolving according to a global, hermitian Hamiltonian [4, 6, 39–60]. We call this the Isolated Quantum System (IQS) approach. A particularly important view within the IQS approach is the so-called Eigenstate Thermalization Hypothesis (ETH) [1–3, 5, 44, 50], which has become a cornerstone in the present understanding of chaos and integrability in quantum many-body systems. In the ETH approach, the initial state of the global set-up is taken to be a generic state which is sharply peaked at some given energy, and it is assumed that the

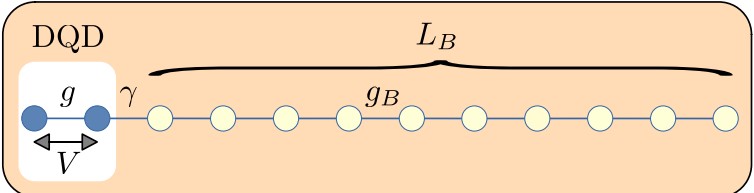

Figure 1:  *Schematic of DQD coupled to a fermionic lead.* The system of interest is that of two fermionic sites (blue discs) which form a double quantum dot (DQD). The DQD is linearly coupled to a chain of $L_B$ fermionic sites (yellow discs) that represent a fermionic lead (bath). The hopping (blue lines) within the lead is $g_B$. The hopping constant within the DQD is $g$, while between the DQD and the lead it is $\gamma$. In addition, a nearest neighbour repulsive many-body interaction with strength $V$ (grey arrow) may act between the two sites of the DQD. The Hamiltonian for the full, closed, spinless fermion chain (orange box) is given by (1). We refer to $V = 0$ as the *free fermion model*, and to $V > 0$ as the *interacting DQD model*.

global Hamiltonian is non-integrable [50]. It is then argued that for large times, the state of any small subsystem of the entire set-up relaxes to the marginal of the global Gibbs state, up to sub-extensive corrections. The temperature of the Gibbs state is consistent with the energy of the global state.

The non-integrability condition is not fulfilled for free or non-interacting models, for which the Hamiltonian is quadratic in bosonic or fermionic creation and annihilation operators. For these models, the ETH approach is known to fail [50, 52]. Yet, the convergence to the MGS has been proven analytically in the OQS approach precisely for this class of Hamiltonians [10–13]. This provides a clear example that the two ways of describing thermalization refer to different physics, although, to our knowledge, a fair comparison between them has never been attempted.

**Summary of results.**  To carry out this case study, we consider the physical model of a double quantum dot (DQD) coupled to a fermionic lead. The model is graphically illustrated in Figure 1. We present here an outline of our results, also summarised in Table 1, and refer to a more detailed discussion in the coming sections.

– For the DQD+lead model, we construct typical initial states $|\psi(0)\rangle$ (details in Section 3.1), which allows exploring both, the open quantum system (OQS) and the isolated quantum system (IQS) notions for thermalization.

– As a first result, we identify a time scale $t_{\text{oqs}}$, proportional to the size of the bath $L_B \gg 1$, before which thermalization can take place according to the OQS approach; see Section 3.2.

– Secondly, we confirm numerically that thermalization according to the OQS approach does take place on the time scale $t_{\text{oqs}}$, the free fermionic case ($V = 0$) and the interacting case ($V \neq 0$); see Section 3.3.

– Thirdly, we compute the spectral form factor (SFF) and the level-spacing distribution $P(s)$ for the whole system to establish the degree of non-integrability (Subsection 3.4). While for $V = 0$ we confirm that the system is integrable, for the interacting DQD ($V = g > 0$) the level distribution changes to a Brody distribution and the SFF behaves to a good approximation as it would be expected for a non-integrable system. We therefore conclude that the interacting DQD is a partially non-integrable model, which we denote as *non-integrable*[*].

– Finally, we obtain numerical results for the dynamics at times $t \gg t_{\text{oqs}}$ (Section 3.5), showing that thermalization in the IQS sense is indeed obeyed for the interacting case. As expected, no such IQS thermalization occurs in the free fermionic case, contrasting with the OQS thermalization found at earlier times. These results demonstrate that both the OQS and

| | | |
|---|---|---|
| DQD+lead Hamiltonian | $\hat{H}(L_B) = \underbrace{g\,(\hat{c}_1^\dagger \hat{c}_2 + \hat{c}_2^\dagger \hat{c}_1) + V\,\hat{n}_1\hat{n}_2}_{\hat{H}_S \text{ [DQD w. interaction } V]} + \underbrace{\gamma\,(\hat{c}_2^\dagger \hat{b}_1 + \hat{b}_1^\dagger \hat{c}_2)}_{\hat{H}_{SB}\text{ [DQD-lead coupling]}} + \underbrace{g_B \sum_{\ell=1}^{L_B-1} (\hat{b}_\ell^\dagger \hat{b}_{\ell+1} + \hat{b}_{\ell+1}^\dagger \hat{b}_\ell)}_{\hat{H}_B(L_B)\text{ [fermionic lead (bath)]}}$ | |
| DQD+lead initial state | $\|\psi(0)\rangle = \dfrac{e^{-\beta \hat{H}^{\text{ini}}(L_B)/2}}{\sqrt{Z^{\text{ini}}(L_B)}} \displaystyle\sum_{n=1}^{D} (a_n + ib_n)\,\|\chi_n\rangle$ | with initial decoupled Hamiltonian $\hat{H}^{\text{ini}}(L_B) = \hat{H}_S^{\text{ini}} + \hat{H}_B(L_B)$ and random Gaussian distributed weights $a_n, b_n$ for basis states $\|\chi_n\rangle$ |
| different regimes | **OQS regime** <br> $t \lesssim t_{\text{oqs}}$ ① with $t_{\text{oqs}} = \dfrac{L_B - L_0}{g_B}$ | **IQS regime** <br> $t \gg t_{\text{oqs}}$ |
| thermalization statement | $\displaystyle\lim_{L_B \to \infty} \hat{\rho}_S(t_{\text{oqs}}; L_B) = \hat{\rho}_{\text{MGS}}$ for reduced system state $\hat{\rho}_S$ | $\overline{\langle \hat{A}\rangle} = A(E) + \mathcal{O}(L_B^{-1})$ <br> $\overline{\delta\langle \hat{A}\rangle^2} \leq C\,e^{-\alpha L_B}$ — for time-averaged expectation value and variance of local observables $\hat{A}$ |
| $V = 0$ (free fermions) | thermalization statement **satisfied** ② | integrable ③ — thermalization statement **not** satisfied ④ |
| $V > 0$ (interacting case) | thermalization statement **satisfied** | **non-integrable\*** — thermalization statement **satisfied** |

Table 1: *Summary of results.*

IQS thermalization statements can be assessed for the DQD coupled to the fermionic lead, putting the emphasis on the crucial role of the time-scale separation and the order of limits. For times $t \lesssim t_{\text{oqs}} \propto L_B$ the DQD follows the thermalization process described by the OQS approach. For much longer times $t \gg t_{\text{oqs}}$ thermalization happens as characterized by the IQS statements, provided the system is not integrable.

## 2 Setting and OQS vs. IQS approaches to thermalization

In this section, we introduce the specific physical model considered here as an illustrative example, as well as the general open quantum system (OQS) and isolated quantum systems (IQS) approaches to describe the dynamics of a quantum system.

### 2.1 Double quantum dot (DQD) coupled to fermionic lead

We consider the dynamics of a DQD coupled to a fermionic chain of length $L_B$, see Fig. 1. The global Hamiltonian is

$$\hat{H}(L_B) = \hat{H}_S + \hat{H}_{SB} + \hat{H}_B(L_B), \tag{1}$$

where

$$\hat{H}_S = g\,(\hat{c}_1^\dagger \hat{c}_2 + \hat{c}_2^\dagger \hat{c}_1) + V\,\hat{n}_1\hat{n}_2, \tag{2}$$

$$\hat{H}_{SB} = \gamma\,(\hat{c}_2^\dagger \hat{b}_1 + \hat{b}_1^\dagger \hat{c}_2), \tag{3}$$

$$\hat{H}_B(L_B) = g_B \sum_{\ell=1}^{L_B-1} (\hat{b}_\ell^\dagger \hat{b}_{\ell+1} + \hat{b}_{\ell+1}^\dagger \hat{b}_\ell). \tag{4}$$

The operator $\hat{H}_S$ describes the Hamiltonian of the bare DQD, the 'system', which consists of two fermionic sites representing two quantum dots. Here $\hat{c}_p$ for $p = 1, 2$ is the fermionic annihilation operator of the $p$-th site, and $\hat{n}_p = \hat{c}_p^\dagger \hat{c}_p$ are the fermionic occupation number operators. The hopping amplitude between the two quantum dots is given by $g$, while $V > 0$ gives the repulsive many-body interaction between the two dots.

The second site of the DQD is coupled to a 'bath': a chain of nearest neighbour tight-binding fermions with $L_B$ sites. The bath Hamiltonian is $\hat{H}_B(L_B)$, with intra-bath hopping constant $g_B$. Here $\hat{b}_\ell$ denotes the fermionic annihilation operator of the $\ell$-th site of the bath. The coupling between the system and the bath is given by the interaction operator $\hat{H}_{SB}$, which is just a hopping term between the second site of the system and the first site of the bath, with hopping strength $\gamma$.

Apart from the $V$-repulsion term in the DQD, all other terms in $\hat{H}(L_B)$ are quadratic in fermion creation and annihilation operators. Therefore, $V = 0$ corresponds to free fermions. However, when $V \neq 0$, then a fourth order many-body interaction term is present in the Hamiltonian, and we will refer to this case as the *interacting DQD*. In the following, we discuss the notion of thermalization in both open quantum system (OQS) and isolated quantum system (IQS) approaches, highlighting their differences, and pointing out how our setting can correspond to both.

## 2.2 OQS thermalization

This section provides general background on the OQS approach applicable for any system-bath Hamiltonian [23–26]. Consider an initial system-bath state of product form $\hat{\rho}_S(0) \frac{e^{-\beta \hat{H}_B}}{Z_B}$, in which the system is in any state $\hat{\rho}_S(0)$, and the bath is in a thermal equilibrium state with respect to the bath's bare Hamiltonian $\hat{H}_B(L_B)$ at inverse temperature $\beta$, with bath partition function $Z_B = \text{Tr}_B\left(e^{-\beta \hat{H}_B}\right)$. Since our setting has conservation of number of particles, a more general initial state would consider a chemical potential for the bath. In this paper, for simplicity, we set the chemical potential to zero. The dynamics of the DQD is obtained by tracing out the bath degrees of freedom of the full density matrix at time $t$,

$$\hat{\rho}_S(t; L_B) = \text{Tr}_B \left[ e^{-it\hat{H}(L_B)} \hat{\rho}_S(0) \frac{e^{-\beta \hat{H}_B(L_B)}}{Z_B} e^{it\hat{H}(L_B)} \right], \tag{5}$$

where we set $\hbar = 1$. The DQD state in the thermodynamic limit, $L_B \to \infty$, is denoted by

$$\hat{\rho}_S(t) = \hat{\Lambda}(t) \left[ \hat{\rho}_S(0) \right] = \lim_{L_B \to \infty} \hat{\rho}_S(t; L_B), \tag{6}$$

which defines $\hat{\Lambda}(t)$, a completely positive trace preserving (CPTP) map on system density matrices. All standard techniques in OQS theory, like quantum master equations, non-equilibrium Green's functions, the Feynman Vernon influence functional, Keldysh-Schwinger path integrals, quantum Langevin equations etc. [23,25,26,28], are formalisms to describe the dynamics of $\hat{\rho}_S(t)$ under this CPTP map. For all these OQS techniques, taking the limit $L_B \to \infty$ is crucial, because it is in this limit that the bath becomes a continuum of fermionic modes and the dynamics becomes dissipative (irreversible).

We now discuss the thermodynamic limit $L_B \to \infty$ for the bath Hamiltonian $\hat{H}_B(L_B)$, (4). It is useful to go to the single particle eigenbasis of the bath by diagonalizing the tight-binding

chain. The bath Hamiltonian and the system-bath coupling Hamiltonian then become

$$\hat{H}_B = \sum_{r=1}^{L_B} \Omega_r \hat{B}_r^\dagger \hat{B}_r \text{ with } \Omega_r = 2g_B \cos\left(\frac{\pi r}{L_B + 1}\right) \text{ and } \hat{H}_{SB} = \sum_{r=1}^{L_B} \kappa_r \left(\hat{c}_2^\dagger \hat{B}_r + \hat{B}_r^\dagger \hat{c}_2\right), \quad (7)$$

$$\text{where } \hat{B}_r = \sum_{\ell=1}^{L_B} \Phi_{\ell r} \hat{b}_\ell \text{ with } \Phi_{\ell r} = \sqrt{\frac{2}{L_B + 1}} \sin\left(\frac{\pi \ell r}{L_B + 1}\right) \text{ and } \kappa_r = \gamma \Phi_{1r}. \quad (8)$$

In the above, $\hat{B}_r$ is the annihilation operator for the $r$-th mode of the bath. It is a standard fact from OQS theory [23, 25, 26, 28] that for such a linearly coupled system and in the $L_B \to \infty$ limit, the influence of the bath on the system dynamics (DQD) is entirely governed by the bath spectral function, also called the hybridization function, defined as

$$\mathfrak{J}(\omega) = 2\pi \lim_{L_B \to \infty} \sum_{r=1}^{L_B} |\kappa_r|^2 \delta(\omega - \Omega_r). \quad (9)$$

In other words, $\hat{\Lambda}(t)$ depends on the bath Hamiltonian parameters only through the quantity $\mathfrak{J}(\omega)$. Using Eq. (7) and (8) in Eq. (9), and converting the summation to an integral taking $q = (\pi r)/(L_B + 1)$ and $dq = \pi/(L_B + 1)$, the bath spectral function for (7) is found to be

$$\mathfrak{J}(\omega) = 4\gamma^2 \int_0^\pi dq \, \sin^2(q) \, \delta(\omega - 2g_B \cos q) = \frac{2\gamma^2}{g_B} \sqrt{1 - \left(\frac{\omega}{2g_B}\right)^2} \quad \text{for } |\omega| \leq 2g_B, \quad (10)$$

and $\mathfrak{J}(\omega) = 0$ for $|\omega| > 2g_B$.

Typically, due to the contact with the bath, the evolution of the system is driven to a stationary state, or steady state, in the long time limit, provided that the system-bath interaction is 'effective', allowing for energy exchange processes between the two components, and the bath is in the thermodynamic limit. This is usually guaranteed to happen if the bath spectral function does not vanish at the Bohr (transition) frequencies of the system, ensuring that bath quanta can induce transitions in the system by absorption and emission processes. This is a general feature of open systems, discussed for example in [12, 16, 17, 20–24]. In such situations, the long-time steady state is very often unique, i.e, independent of the initial state of the system. We will only consider parameter regimes where there is a unique steady state.

Given this setting, the statement of thermalization in the OQS approach is:

$$\lim_{t \to \infty} \left(\lim_{L_B \to \infty} \hat{\rho}_S(t; L_B)\right) = \hat{\rho}_{\text{MGS}}, \quad (11)$$

where we recall the definition of the reduced density matrix $\hat{\rho}_S(t; L_B)$ in (5), and where the *mean force Gibbs state* is defined by

$$\hat{\rho}_{\text{MGS}} = \lim_{L_B \to \infty} \text{Tr}_B \left[\frac{e^{-\beta \hat{H}(L_B)}}{Z(L_B)}\right], \quad (12)$$

with total partition function $Z(L_B) = \text{Tr}\left[e^{-\beta \hat{H}(L_B)}\right]$. In other words, the state of the system should converge to the mean force Gibbs state in the long time and thermodynamic limits of the set-up, irrespective of the initial state of the system. However, it is important to note the order of limits. In the OQS approach, the system is expected to converge to its asymptotic, mean force Gibbs state if *first* one takes the large bath limit, $L_B \to \infty$, and *afterwards* one takes the long time limit, $t \to \infty$.

We note that although the above statement of OQS thermalization is expected to hold generally, so far it has only been proven for special cases. These include free fermionic and free bosonic models [7–13] and a range of results in the regime of ultraweak and weak coupling between system and bath [7, 9, 18–23, 36, 38, 61–66] Hence it becomes important to numerically check the notion of thermalization of OQS beyond weak system-bath coupling and with non-linear interactions present, where analytical proofs that OQS occurs are lacking. The task will thus be to check for the validity of Eq. (11) for the DQD in the presence of the non-linear $V$ interaction, see Fig. 1.

## 2.3 IQS thermalization

Now turning to the isolated quantum system (IQS) approach, here one does not make a distinction between the DQD and the fermionic lead. Rather, one considers the two together as an isolated system. Now, an observable $\hat{A}$, that is, in our case, an operator acting on the whole DQD plus fermionic lead, is called *local* if it acts non-trivially on a fixed set of sites, independently of how large we take $L_B$, or is a sum of such observables. For example, observables acting on the first two sites (on the DQD, see. Fig. 1) are local.

To state the thermalization in the IQS approach, we look at the expectation value of the operator $\hat{A}$ as a function of time

$$\langle \hat{A}(t) \rangle = \langle \psi | e^{i\hat{H}t} \hat{A} e^{-i\hat{H}t} | \psi \rangle. \tag{13}$$

The initial state $|\psi\rangle$ is supposed to have a small energy spread around the given value $E = \langle \psi | \hat{H} | \psi \rangle$. This is usually quantified as

$$\frac{\langle \psi | \hat{H}^2 | \psi \rangle - \langle \psi | \hat{H} | \psi \rangle^2}{\langle \psi | \hat{H} | \psi \rangle^2} \sim O(L_B^{-1}). \tag{14}$$

We denote by $A(E)$ the expectation value of $\hat{A}$ in the microcanonical ensemble at energy $E$ [67]. From the equivalence of ensembles in standard statistical physics, one has

$$A(E) = \lim_{L_B \to \infty} \mathrm{Tr} \left[ \hat{A} \, \frac{e^{-\beta_E \hat{H}}}{Z} \right], \tag{15}$$

where the microcanonical entropy $S$ at energy $E$ is used to define the microcanonical inverse temperature,

$$\beta_E = \frac{\partial S(E)}{\partial E}. \tag{16}$$

Additionally, since we have conservation of number of particles, we should have chemical potential. In the microcanonical picture, setting the chemical potential to zero corresponds to a half-filling of the entire chain. Convergence to the grand canonical ensemble with zero chemical potential is guaranteed if the state $|\psi\rangle$ is sharply peaked around the half-filled sector.

In this setting, the statement of thermalization in the IQS approach now is [4, 39–46, 49–56]

$$\overline{\langle \hat{A} \rangle} := \lim_{t \to \infty} \frac{1}{t} \int_0^t dt' \langle \hat{A}(t') \rangle = A(E) + \mathcal{O}(L_B^{-1}), \tag{17}$$

$$\overline{\delta \langle \hat{A} \rangle^2} := \lim_{t \to \infty} \frac{1}{t} \int_0^t dt' \left| \langle \hat{A}(t') \rangle - \overline{\langle \hat{A} \rangle} \right|^2 \leq C e^{-\alpha L_B}. \tag{18}$$

The first equalities in (17) and (18) are definitions. The second equality in (17) and the inequality in (18) defines the meaning of thermalization in IQS. In Eq. (18), the constants $C$ and $\alpha$ are non-universal positive constants which are independent of $L_B$.

Equation (17) says that the infinite time average of the expectation value deviates from its value obtained from the microcanonical ensemble by a small term, having size at most of the order of $L_B^{-1}$. The second IQS thermalization statement (18) says that time-averaged fluctuations about the infinite time average are exponentially small in $L_B$, in the long time limit. These two statements combined say that for large times, the time-averaged expectation value of the local observable approaches its (infinite time) mean value modulo exponentially small fluctuations in $L_B$.

For large enough values of $L_B$, and since, in our example, the DQD consists of only the first two sites of chain (see. Fig. 1), the inverse temperature $\beta_E$ in Eq. (16) depends minimally on the initial state of the DQD, and equals to a very good approximation the fermionic lead's inverse temperature $\beta$, governed by the lead's initial state. Then, for local operators $\hat{A}_S$ that only act on the DQD Hilbert space, the IQS statement of thermalization given in Eq.(17) becomes

$$\overline{\langle \hat{A}_S \rangle} = A_{\mathrm{MGS}} + \mathcal{O}(L_B^{-1}), \tag{19}$$

where $A_{\mathrm{MGS}}$, defined as

$$A_{\mathrm{MGS}} = \mathrm{Tr}\left[ \hat{A}_S \, \hat{\rho}_{\mathrm{MGS}} \right], \tag{20}$$

is the expectation value of $\hat{A}_S$ in the mean force Gibbs state (12). The inverse temperature $\beta$ appearing in the state $\rho_{\mathrm{MGS}}$ is the inverse temperature obtained from initial state of the fermionic lead. Due to the small contribution to the total energy stemming from the interaction between the DQD and the fermionic lead, beyond a minimal lead size $L_B$, the value of $A_{\mathrm{MGS}}$ will remain approximately constant under further increase of $L_B$.

Considering local observables on the DQD, and combining Eq. (19) with Eqs. (17) and (18), we see that thermalization in the IQS approach also means convergence of the state of the DQD to $\hat{\rho}_{\mathrm{MGS}}$, in the long time and thermodynamic limit. At first sight this appears similar to thermalization in the OQS approach. However, there are crucial differences, which we describe below.

## 2.4 Distinguishing OQS and IQS thermalization

The main difference between the OQS and IQS notions of thermalization is the order in which the long time and the thermodynamic limits are taken. The OQS thermalization refers to convergence to $\hat{\rho}_{\mathrm{MGS}}$ when the thermodynamic limit ($L_B \to \infty$) is taken first, and then the long-time limit ($t \to \infty$) is taken (see Eq. (11)). In the IQS approach, first an infinite time average of any observable is taken for a fixed finite $L_B$. The statements of the IQS thermalization tell us how this value approaches the corresponding value obtained from $\hat{\rho}_{\mathrm{MGS}}$, and how the fluctuations about the latter decay, with increasing $L_B$ (see Eqs.(17),(18)).

This difference has important consequences. In most of the literature, thermalization in the IQS approach is discussed for quantum many-body systems, i.e, in the presence of many-body interactions [4, 39–60]. In particular, the most well-accepted mechanism for IQS thermalization is the so-called eigenstate thermalization hypothesis (ETH) [1–3, 5, 44, 50], which is expected to apply to non-integrable quantum systems. Thermalization in the IQS has been numerically established in a range of quantum non-integrable many-body systems [4, 44, 47, 48, 50, 52, 53, 55, 56, 58, 60]. The requirement of non-integrability excludes cases such as free fermions and free bosons, where many-body interaction terms are absent in the Hamiltonian. ETH does not apply and IQS thermalization is not expected to occur in such systems. This is in stark contrast with the fact that proofs for the OQS thermalization are known only for free fermionic and free bosonic systems [7–13, 16–22]. Thus it may appear that the two notions of thermalization are at odds with each other, predicting that the same system does or does not thermalize. In the current work, we resolve this seeming inconsistency.

Recalling the concepts of OQS and IQS thermalizations discussed in Secs. 2.2 and 2.3 above, there seems to be a second key difference: while OQS thermalization is discussed for initial states of the whole set-up that are mixed, IQS thermalization is usually discussed for initial pure states. As we will discuss in Sec. 3.1 below, this difference can be completely bridged. Before we get to that, we next outline how to obtain the state of the DQD as a function of time.

## 2.5 Obtaining the density matrix of the DQD

To present the equilibration results for the OQS and the IQS approach in section 3, we use four local DQD operators which fully describe the DQD state, as we explain now.

For any initial state $\hat{\rho}(0)$ of the DQD plus the fermionic leads, see Eq. (5), the reduced DQD density matrix at time $t$ is given by

$$\hat{\rho}_S(t; L_B) = \mathrm{Tr}_B \left[ e^{-it\hat{H}(L_B)} \hat{\rho}(0) e^{it\hat{H}(L_B)} \right]. \tag{21}$$

The global Hamiltonian $\hat{H}(L_B)$ is number conserving, i.e., it commutes with the total number operator $\hat{N}_{\mathrm{tot}} = c_1^\dagger c_1 + c_2^\dagger c_2 + \sum_{\ell=1}^{L_B} b_\ell^\dagger b_\ell$. When the initial state $\hat{\rho}(0)$ also commutes with the total number operator

$$[\hat{\rho}(0), \hat{N}_{\mathrm{tot}}] = 0, \tag{22}$$

then the reduced density matrix $\hat{\rho}_S(t; L_B)$ leaves the sectors of a fixed number (0 or 1 or 2) of excitations of the DQD invariant. In this case the time-evolved DQD density matrix is of the form

$$\hat{\rho}_S(t; L_B) = \begin{bmatrix} \rho_{00,00}(t) & 0 & 0 & 0 \\ 0 & \rho_{01,01}(t) & \rho_{01,10}(t) & 0 \\ 0 & \rho_{10,01}(t) & \rho_{10,10}(t) & 0 \\ 0 & 0 & 0 & \rho_{11,11}(t) \end{bmatrix}, \tag{23}$$

written in the ordered orthonormal basis $\{|00\rangle, |01\rangle, |01\rangle, |11\rangle\}$ of the DQD Hilbert space, labeling the excitations of the two DQD sites. Using the notation

$$\langle \hat{A}(t) \rangle := \mathrm{Tr} \left[ \hat{A} \, e^{-it\hat{H}(L_B)} \, \hat{\rho}(0) \, e^{it\hat{H}(L_B)} \right], \tag{24}$$

the matrix elements of (23) are expressed entirely in terms of the expectations of the four DQD operators

$$\langle \hat{n}_1(t) \rangle, \ \langle \hat{n}_2(t) \rangle, \ \langle \hat{c}_1^\dagger(t) \hat{c}_2(t) \rangle, \ \langle \hat{n}_1(t) \hat{n}_2(t) \rangle, \tag{25}$$

which of course depend on the bath size $L_B$. Explicit, we have

$$\begin{aligned} \rho_{01,10}(t) &= \rho_{10,01}^*(t) = \langle \hat{c}_1^\dagger(t) \hat{c}_2(t) \rangle \\ \rho_{11,11}(t) &= \langle \hat{n}_1(t) \hat{n}_2(t) \rangle \\ \rho_{01,01}(t) &= \langle \hat{n}_1(t) \rangle - \langle \hat{n}_1(t) \hat{n}_2(t) \rangle \\ \rho_{10,10}(t) &= \langle \hat{n}_2(t) \rangle - \langle \hat{n}_1(t) \hat{n}_2(t) \rangle \\ \rho_{00,00}(t) &= 1 - \rho_{10,10}(t) - \rho_{01,01}(t) - \rho_{11,11}(t). \end{aligned} \tag{26}$$

In this paper, we will consider states satisfying Eq. (22) for all our analysis, without further mention.

## 3 Results

In this Section we present our main results, which were summarised in Table 1 above.

### 3.1 Treating OQS and IQS on same footing via dynamical typicality

We first explain the concept of dynamical typicality and then we show how we can use that to treat both the OQS and the IQS formalisms on the same footing.

Let $\{|\chi_n\rangle\}_{n=1}^{D_{\text{tot}}}$ be an arbitrary orthonormal basis of the entire system plus bath Hilbert space of dimension $D_{\text{tot}} = 2^{L_B+2}$. Let also $a_n$ and $b_n$ be independent, identically distributed random numbers with Gaussian distribution of mean 0 and variance $1/2$. A state of the form

$$|\psi\rangle = \hat{R} \sum_{n=1}^{D_{\text{tot}}} (a_n + ib_n)|\chi_n\rangle, \tag{27}$$

where $\hat{R}$ is an arbitrary linear operator in the Hilbert space, is called a *typical state* [68, 69]. Denote by $\mathbb{E}$ the average (expectation) with respect to the Gaussian distribution of the $a_n, b_n$. It can be verified that for any observable $\hat{A}$ we have $\mathbb{E}\big[\langle\psi|\hat{A}|\psi\rangle\big] = \text{Tr}\big[\hat{R}\hat{R}^\dagger\hat{A}\big]$ [68,69]. If $\hat{R}$ is chosen such that $\hat{R}\hat{R}^\dagger$ is a density matrix, then the last equality means that the expectation value of operator $\hat{A}$ averaged over the ensemble of pure states $|\psi\rangle$ is the same as taking the quantum-mechanical average of $\hat{A}$ in the state $\hat{\rho} = \hat{R}\hat{R}^\dagger$. Moreover, if $\text{Tr}[\hat{\rho}^2] \ll 1$, i.e, if the density matrix $\hat{\rho}$ is sufficiently mixed, then the sample to sample fluctuations coming from the Gaussian distribution of the $a_n, b_n$, decay as $\sim D_{\text{tot}}^{-1}$ [68, 69]. Since, in the current model, $D_{\text{tot}} = 2^{L_B+2}$, those fluctuations are exponentially suppressed with increase in the lattice size $L_B$. Therefore, for large $L_B$, operator expectation values obtained from a *single realization* of the typical state $|\psi\rangle$ closely approximate those obtained from the ensemble averaged state $\hat{\rho} = \hat{R}\hat{R}^\dagger$,

$$\langle\psi|\hat{A}|\psi\rangle \sim \text{Tr}\big[\hat{\rho}\hat{A}\big]. \tag{28}$$

Here the $\sim$ sign means that a single realization gives an operator expectation value very close to that obtained for the Gaussian averaged state. This is the statement of *dynamical typicality* [68, 69].

Our main idea here is to use dynamical typicality to pass between pure initial states used in the formulation of the IQS approach (Section 2.3) and mixed (equilibrium) initial states used in the OQS approach (Section 2.2). We will therefore use, for both the OQS and IQS approaches, initial states of the form

$$|\psi(0)\rangle = \frac{e^{-\beta\hat{H}^{\text{ini}}(L_B)/2}}{\sqrt{Z^{\text{ini}}(L_B)}} \sum_{n=1}^{D_{\text{tot}}} (a_n + ib_n)|\chi_n\rangle, \tag{29}$$

where $Z^{\text{ini}}(L_B)$ is a normalization constant. The operator in front of the sum in (29) is the square root of the thermal density matrix associated with the uncoupled system-bath Hamiltonian

$$\hat{H}^{\text{ini}}(L_B) = \hat{H}_S^{\text{ini}} + \hat{H}_B(L_B), \tag{30}$$

where the fermionic lead Hamiltonian $\hat{H}_B(L_B)$ is given in (4). Furthermore, $\hat{H}_S^{\text{ini}}$ is a new initial-state system-Hamiltonian, defined by

$$\hat{H}_S^{\text{ini}} = \varepsilon_1^{\text{ini}}\,\hat{n}_1 + \varepsilon_2^{\text{ini}}\,\hat{n}_2 + g^{\text{ini}}\left(e^{i\phi^{\text{ini}}}\,\hat{c}_1^\dagger\hat{c}_2 + e^{-i\phi^{\text{ini}}}\,\hat{c}_2^\dagger\hat{c}_1\right) + V^{\text{ini}}\hat{n}_1\hat{n}_2, \tag{31}$$

where the parameters $\varepsilon_{1,2}^{\text{ini}}$, $g^{\text{ini}}$, $\phi^{\text{ini}}$ and $V^{\text{ini}}$ are arbitrary real numbers.

According to (28), applied to the observable $\hat{A}(t) = e^{it\hat{H}(L_B)}\hat{A}e^{-it\hat{H}(L_B)}$, the dynamics of expectation values starting in the pure initial state in Eq. (29) (for any randomly chosen values of $a_n, b_n$ and $L_B$ large) is indistinguishable from the dynamics starting in the total mixed state

$$\hat{\rho}(0) = \frac{e^{-\beta\hat{H}^{\text{ini}}(L_B)}}{Z^{\text{ini}}(L_B)} = \hat{\rho}_S(0)\,\frac{e^{-\beta\hat{H}_B(L_B)}}{Z_B(L_B)} \tag{32}$$

where $Z^{\text{ini}}(L_B) = Z_S^{\text{ini}} Z_B(L_B)$ and the initial system state is expressed as $\hat{\rho}_S(0) = \frac{e^{-\beta \hat{H}_S^{\text{ini}}}}{Z_S^{\text{ini}}}$. The initial state $\hat{\rho}(0)$ is thus a product of the DQD and the fermionic lead, where the lead always starts in a thermal state at inverse temperature $\beta$. Varying over the parameters in $\hat{H}_S^{\text{ini}}$ allows us to describe *arbitrary* initial states of the DQD, which furthermore commute with the number of particles in the DQD. Note that $\hat{\rho}(0)$ commutes with the operator of total number of excitations, $N_{\text{tot}}$ (see also(22)).

Finally we note that $\hat{\rho}(0)$ in Eq.(32) is a thermal state of a short range Hamiltonian, which differs from $\hat{H}(L_B)$ only in the first two sites. By virtue of equivalence of ensembles of statistical physics, this guarantees that $\hat{\rho}(0)$ has a small spread in energy. The analog of Eq. (14) is then satisfied for the expectation values for the mixed state $\hat{\rho}(0)$ obtained from the Hamiltonian $\hat{H}^{\text{ini}}(L_B)$. The same will hold when calculating the expectation values with respect to $\hat{H}(L_B)$ instead. Since this is the same as obtaining the expectation values from $|\psi(0)\rangle$ in Eq. (29), our choice of initial state satisfies Eq. (14). We have also confirmed this numerically.

Thus, for exploring both OQS and IQS thermalization, we need to evolve the state $|\psi(0)\rangle$ in Eq. (29) to a long time using the full Hamiltonian of DQD and the fermionic lead, for finite but large enough $L_B$. We need to obtain the four DQD expectation values given in Eq. (25) as a function of time. Obtaining such data for various values of $L_B$ allows us to check both OQS thermalization [Eq. (11)] and IQS thermalization [Eqs. (17), (18), (19)]. We require neither separate models nor separate initial states to consider thermalization in OQS and IQS approaches. Rather, as we establish later, we only need to consider separate time regimes of the same data obtained from the same model starting with the same initial state. This allows a fair comparison of the two approaches to thermalization and highlights the difference between the two.

**Numerical technique.** Before we present our numerical results, we here briefly comment on how the numerics were carried out. As mentioned in Sections 2.2 and 2.3, for both the OQS and IQS approaches, one must consider large bath sizes $L_B$ and simulate the dynamics up to long enough times. For $V \neq 0$, free-fermion techniques do not apply and numerical implementations are hard because the Hilbert space dimension $D$ scales exponentially with $L_B$. Nevertheless, the global Hamiltonian Eq. (1) describes a nearest neighbour tight-binding chain of spinless fermions, with nearest neighbour many-body interaction only between the first two sites. Written in the Fock state basis, the Hamiltonian is block diagonal because number of particles is conserved. Additionally, it is a sparse matrix. Within our computational resources, exact diagonalization is possible up to $L_B = 18$ (total chain length is 20 sites). However, this size is not large enough to clearly demonstrate all of the effects we wish to explore. We thus use the Chebyshev polynomial method [70–73], in which the dynamics can be simulated without diagonalizing the Hamiltonian, but rather repeatedly using sparse matrix vector multiplications. The initial state in Eq. (29) can also be prepared similarly, using imaginary time evolution. With this method, we can reach up to $L_B = 26$ within our computational resources (total chain length is 28 sites, size of the largest block of Hamiltonian in Fock space is $40,116,600$). More importantly, it allows the simulation up to extremely long times (up to $\sim 2000gt$). The chain sizes and times treatable with this method are large enough to clearly explore both OQS and IQS regimes.

## 3.2 Relevant time-scale in the OQS approach

We now discuss the equilibration dynamics of the DQQ in contact with the fermionic lead, within the open quantum systems approach.

The fermionic lead plays the role of the bath for the DQD. This bath is a nearest neighbour tight-binding chain and only the first site of the bath is coupled to the DQD, see Fig. 1. At

the initial time, the system-bath coupling is switched on. Due to the bath being a short-ranged lattice system, Lieb-Robinson bounds dictate that, at any finite time $t$, only a finite part of the bath is affected by this switching-on. The 'disturbance' given by the switching-on of system-bath coupling spreads with the Fermi velocity, $v_F = 2g_B$ and reaches the site $L_B$ in time $t_1 \simeq L_B/(2g_B)$. Then, it is reflected back. The DQD is influenced by the finite size of the bath when this reflected 'signal' reaches it. The reflected 'signal' again travels with Fermi velocity, so, the dynamics of the system is negligibly affected by the finiteness of the tight-binding chain representing the bath till $t \simeq 2t_1 \simeq L_B/g_B$. In practice, therefore, to exactly obtain the dynamics of the system in the presence of a macroscopic bath up to a time $t$, it suffices to consider a bath of size

$$L_B = g_B t + L_0, \tag{33}$$

where $L_0$ is a chosen fixed offset, independent of $t$ and $L_B$. The fixed offset $L_0$ is chosen such that there is sufficiently small finite bath-size effect. We numerically find it sufficient to choose $L_0 = 8$. So, for our setting, the statement of thermalization in OQS approach,i.e, Eq. (11), can be recast as

$$\lim_{t \to \infty} \hat{\rho}_S(t; g_B t + L_0) = \lim_{L_B \to \infty} \hat{\rho}_S \left( \frac{L_B - L_0}{g_B}; L_B \right) = \hat{\rho}_{\mathrm{MGS}}. \tag{34}$$

In particular, in the second equality above, the functional dependence of time on $L_B$ ensures that the time is not long enough to resolve the discreteness of the bath energy levels, and the DQD perceives the bath as a continuum. Combining all the discussion in this section and in Sec. 3.1, we see that, for our choice of initial state Eq.(29), for a given $L_B$ and a chosen fixed $L_0$, the dynamics of the DQD up to time

$$t_{\mathrm{oqs}} = \frac{L_B - L_0}{g_B} \tag{35}$$

corresponds to the OQS regime. In other words, for a finite bath of size $L_B$, the time $t_{\mathrm{oqs}}$ is the relevant timescale up to which the open system approach is suitable for describing equilibration. This is the first result of the paper.

Thermalization of the DQD in the OQS approach then corresponds to convergence of the state of the DQD to $\hat{\rho}_{\mathrm{MGS}}$ within this regime, starting from a global initial pure state of the form given in Eq. (29). We re-iterate that, in this choice of initial state, the arbitrariness of the parameters in $\hat{H}_S^{\mathrm{ini}}$ makes the initial state of the DQD arbitrary. The convergence should happen irrespective of the DQD initial state, i.e, irrespective of the choice of parameters in $\hat{H}_S^{\mathrm{ini}}$. Such thermalization is expected to hold both for free fermion case ($V = 0$), and the interacting DQD case ($V \neq 0$). This is indeed the case, as we show numerically below.

## 3.3 Numerical results for the OQS approach

Our main numerical results are given in Fig. 2. In particular, graphs (a) and (d) illustrate the correctness of Eq. (34), both for the free fermion case ($V = 0$, panels (a)–(c)) and the interacting case ($V \neq 0$, panels (d)–(f)).

To simulate the dynamics starting from an arbitrary, but fixed, initial state of the DQD we choose a set of random parameters in $\hat{H}_S^{\mathrm{ini}}$. Panels (a) and (d) of Fig. 2 show the dynamics of a representative observable $\langle \hat{n}_1(t)\hat{n}_2(t) \rangle$, see Eq. (25), in the regime $t \leq t_{\mathrm{oqs}}$ with $t_{\mathrm{oqs}}(L_B)$ given in (35), for various choices of $L_B$, and one choice of initial state $|\psi(0)\rangle$, see Eq. (29). One clearly sees that plots for smaller values of $L_B$ overlap with those of larger values of $L_B$, demonstrating that there is no finite bath size effect in this regime. With increase in $L_B$, which leads to increase in $t_{\mathrm{oqs}}$, the observable expectation value converges to that obtained from $\hat{\rho}_{\mathrm{MGS}}$. Note,

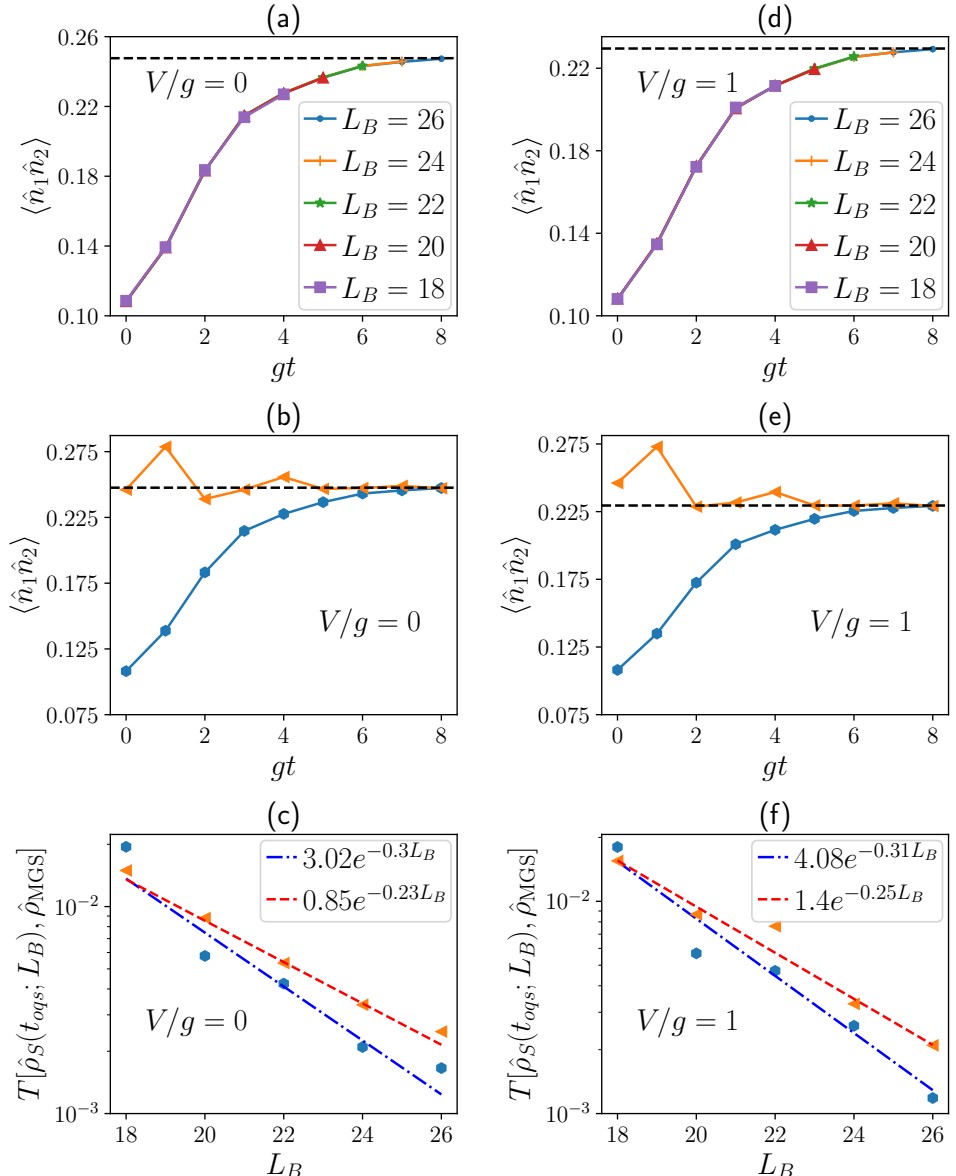

Figure 2: *Thermalization in the OQS regime:* Panels (a), (b), (c) are for the free fermion case, $V = 0$. Panels (d), (e), (f) are for the interacting DQD case with $V = g$. In panels (a) and (d) we plot the expectation value of a representative observable, $\langle \hat{n}_1 \hat{n}_2 \rangle$, as a function of time, for different bath sizes $L_B$, starting from one randomly chosen initial state of the DQD. For each $L_B$, the time regime $t \leq t_{\text{oqs}}$ is considered, which corresponds to the OQS regime. Plots for larger values of $L_B$ overlap with those of smaller values of $L_B$ but extend to longer times. In panels (b) and (e), we plot the same, but for fixed $L_B = 26$ and two different randomly chosen initial states (blue and yellow symbols) of the DQD. The horizontal dashed lines in panels (a), (b), (d), (e) show the corresponding expectation value obtained with $\hat{\rho}_{\text{MGS}}$. In panels (c) and (f), we plot the trace distance between $\hat{\rho}_S(t_{\text{oqs}}; L_B)$ and $\hat{\rho}_{\text{MGS}}$ as a function of bath size $L_B$ for two different choices of initial states (blue and orange symbols) of the DQD. The straight lines (dashed-dotted blue and dashed red) are exponential fits to the symbols, with numerical fit parameters given. Other parameters: $g_B = 2g$, $\gamma = g$, $\beta g = 0.1$, $L_0 = 8$.

that $\hat{\rho}_{\mathrm{MGS}}$ and hence the dashed lines depend on the value of $V$. In panels (b) and (e) of Fig. 2, we display the dynamics of $\langle \hat{n}_1(t)\hat{n}_2(t)\rangle$ in the regime $t \leq t_{\mathrm{oqs}}$ for two different randomly chosen initial states of the DQD, keeping $L_B = 26$. The plots clearly demonstrate that irrespective of its initial value, the average of the observable converges to its value in $\hat{\rho}_{\mathrm{MGS}}$. This is the second result of the paper.

In panels (c) and (f) of Fig. 2, we further plot the trace distance $T$ between $\hat{\rho}_S(t_{\mathrm{oqs}}; L_B)$, the density matrix of the DQD (21) at time $t_{\mathrm{oqs}}$ defined in (35), and the mean force Gibbs state $\hat{\rho}_{\mathrm{MGS}}$, (12) as a function of $L_B$ for two different initial states of the DQD.[1] We find that the trace distance quickly decays with $L_B$, directly evidencing the convergence of the state, without relying on expectation values of observables.

As we will see next, for a given $L_B$, the IQS regime is reached when the evolution is continued beyond time $t_{\mathrm{oqs}}$, where only the interacting DQD case (and not the free fermion case) will show thermalization, in stark contrast with the OQS regime.

## 3.4 Partial non-integrability of the interacting DQD case

Before we proceed to analyse equilibration within the IQS approach, we must distinguish between the integrable and the non-integrable case, since, as mentioned before, IQS thermalization is mainly discussed for non-integrable systems [4, 39–46, 49–60]. It has been established that a single impurity in an otherwise integrable model can make the system non-integrable [74–77]. But, to our knowledge, this has not been checked for the interacting DQD situation that we are considering. We check the integrability of our model using the spectral form factor (SFF), as well as the level-spacing distribution. In order to do so, we restrict ourselves to the half-filled sector of the full Hamiltonian, whose dimension, we denote by $D$. The SFF is defined as

$$\mathrm{SFF} = \frac{1}{D^2}\left|\mathrm{Tr}(e^{-i\hat{H}t})\right|^2. \tag{36}$$

It is known [78–87], that for non-integrable systems the SFF shows a characteristic dip, ramp and plateau behaviour as a function of $t$. The ramp and plateau expression has the form $b_2(Wt/(2\pi D))$, where $W = E_{max} - E_{min}$ is the difference between largest and smallest eigenvalue of $\hat{H}$ and the function $b_2$ is

$$b_2(x) = \frac{1}{D}\left[1 - [1 - 2x + x\log(2x+1)]\,\Theta(1-x) + \left[x\log\left(\frac{2x+1}{2x-1}\right) - 1\right]\Theta(x-1)\right], \tag{37}$$

with $\Theta(x)$ being the Heaviside step function. At $x = 1$ the function switches from an approximately linear ramp, to an approximately constant plateau with the value $1/D$. For integrable systems, this universal ramp and plateau behaviour will be missing.

We numerically evaluate the SFF for both the non-interacting DQD case and the interacting DQD case. The plots of the SFF are shown in blue in Fig. 3. It is clear that the non-interacting DQD case does not show any resemblance to $b_2$ (orange) even at long times. However, the interacting DQD case with $V = g$ indeed shows a ramp and plateau which resembles the universal function $b_2$. This provides numerical evidence that the interacting DQD case has non-integrable features.

Integrable vs. chaotic behaviour can also be assessed by analysing the energetic spectra [50]. For example, the spectra for transmon qubit gates have been found to show chaotic fluctuations [88]. Here we follow this approach and calculate the energetic level spacing distribution for the full Hamiltonian in the half-filled sector. Let us write the full Hamiltonian as $\hat{H} = \sum_{\alpha=1}^{D} E_\alpha |E_\alpha\rangle\langle E_\alpha|$, where $E_\alpha$ are the energy eigenvalues, arranged in ascending order of

---

[1]The trace distance between any two density matrices $\hat{\rho}_1$ and $\hat{\rho}_2$ is defined as $T(\hat{\rho}_1, \hat{\rho}_2) = \frac{1}{2}\sum_{r=1}^{d}|\lambda_r|$, where $\lambda_r$, $r = 1, 2, 3, \ldots d$ are the eigenvalues of $\hat{\rho}_1 - \hat{\rho}_2$, and $d$ is the Hilbert space dimension.

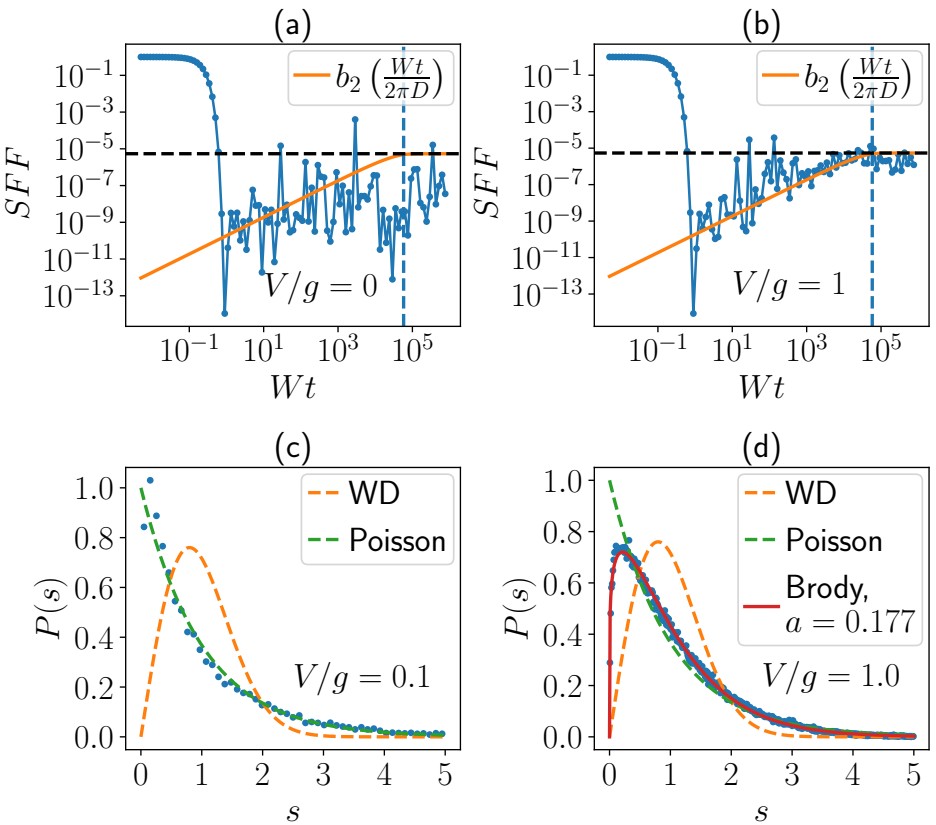

Figure 3: *Checking integrability of the interacting DQD:* The spectral form factor (SFF) (36) of the full Hamiltonian $\hat{H}$ in (1) is plotted (blue symbols) as a function of $Wt$, with $W = E_{max} - E_{min}$, for the non-interacting DQD case $V = 0$ in panel (a), and for the interacting DQD case with $V/g = 1$ in panel (b). The function $b_2$ (orange line) is the universal expression given in Eq. (37). The SFF in panel (a) does not approach the function $b_2$, while in panel (b), after an initial dip, the SFF oscillates around $b_2$, evidencing the SFF behaviour expected of a non-integrable model. The vertical dashed line shows the transition point from ramp to plateau, at $Wt = 2\pi D$, with $D = 2^{L_B+2}$. The results in (a) and (b) are for $L_B = 16$ (total chain size $L_B + 2 = 18$). The horizontal dashed line is located at $1/D$. Panel (c) shows the system's level-spacing distribution $P(s)$ (blue dots) at $V/g = 0.1$ (almost free fermion case) for $L_B = 16$. It is evident that it is close to the Poissonian distribution $P^{\text{Poi}}(s)$, (39) (dashed green), as expected for an integrable system. Panel (d) shows the level distribution $P(s)$ for the interacting DQD model at $V/g = 1.0$ for $L_B = 16$. The level-spacing distribution is still far from the Wigner-Dyson (WD) distribution (dashed yellow) expected for fully non-integrable systems. But, $P(s)$ closely follows a Brody distribution (red line) which, unlike the Poisson distribution (dashed green), goes to zero at zero spacing $s$.

energy, $|E_\alpha\rangle$ are the corresponding energy eigenvectors, and $d$ is the dimension of the half-filled Hilbert space. The level-spacing distribution is given by

$$P(s) = \frac{1}{\mathcal{N}} \sum_{\alpha=1}^{D-1} \delta \left( s - \frac{E_{\alpha+1} - E_\alpha}{\Delta} \right), \tag{38}$$

where $\Delta$ is the mean nearest neighbour level spacing, $\mathcal{N}$ is a normalization constant, and $\delta(x)$ is the Dirac delta function. For integrable systems, a large number of degeneracies of energy

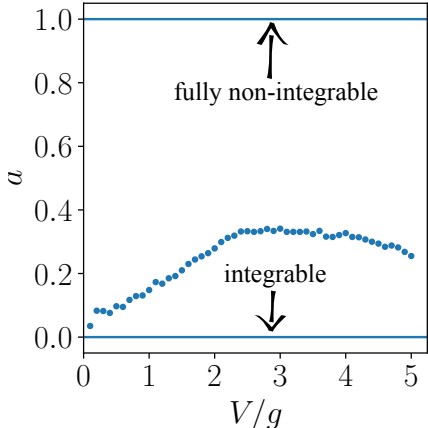

Figure 4: *Partial non-integrability of the interacting DQD:* The figure shows the values of $a$, obtained by fitting the level-spacing distribution at half filling of the interaction DQD with the Brody distribution, as a function of interaction strength $V$. The value $a = 0$ corresponds to a Poisson distribution, which is expected for integrable systems, while $a = 1$ corresponds to a Wigner-Dyson distribution, expected for fully non-integrable systems. We see that even on increasing $V$, $a$ does not approach 1. This evidences non-integrable* behaviour of the interacting DQD. The results are for $L_B = 16$ (total chain length $L_B + 2 = 18$).

levels is expected. The corresponding level spacing distribution is Poissonian [47, 48, 50, 77]

$$P^{\text{Poi}}(s) \propto e^{-s} \qquad \text{(for integrable systems)}. \tag{39}$$

For non-integrable systems, no degeneracy of energy levels within a symmetry sector is expected. This is termed level-repulsion. In fact, if our system is fully non-integrable, the level-spacing distribution is expected to correspond to that of a random matrix of the Gaussian orthogonal ensemble (GOE), which is called the Wigner-Dyson (WD) distribution [47, 48, 50, 77],

$$P^{\text{WD}}(s) = \frac{\pi s}{2} e^{-\pi s^2/4} \qquad \text{(for fully non-integrable systems)}. \tag{40}$$

There is another distribution called the Brody distribution that smoothly interpolates between the Poisson distribution and the WD distribution [47, 48, 77],

$$P^{\text{Bro}}(s) = (a+1) \, b \, s \, e^{-b \, s^{(a+1)}} \quad 0 \le a \le 1 \quad \text{(for partially non-integrable systems)}, \tag{41}$$

where $b = \Gamma\left(\frac{a+1}{a+1}\right)^{a+1}$. For $a = 0$, the Brody distribution becomes the Poisson distribution, while for $a = 1$, it becomes the WD distribution. Importantly, the Brody distribution for $a > 0$ also goes to zero at $s = 0$, signifying level repulsion.

Remarkably, we find that the interacting DQD Hamiltonian satisfies the Brody distribution with $0 < a < 1$. The spacing distribution for our system is shown in Figs. 3(c) and (d). In Fig. 3(c), we show the spacing distribution for a small value of $V/g$. For such small values of $V/g$, at accessible system sizes, the distribution is close to $P^{\text{Poi}}(s)$. However, on increasing $V/g$, the level repulsion becomes clear, and the distribution fits very well to a Brody distribution with $0 < a < 1$. This is shown for $V/g = 1$ in Fig. 3(d). In Fig. 4, we show plot of the Brody-distribution parameter $a$ versus the interaction strength. We find that the WD distribution is not recovered even for larger values of $V/g$. Clearly, this system is neither integrable nor fully non-integrable. We call this system 'partially non-integrable', denoted in short as non-integrable*. Identifying the non-integrable* nature of the interacting DQD is the third result of the paper.

Level spacing distributions corresponding to Brody distributions have been previously explored to characterize onset of quantum chaos [47,48,77]. In such cases, on sufficiently increasing a integrability-breaking parameter, the WD distribution is recovered. The fact that, in our case, the WD distribution is not recovered despite increasing the integrability breaking parameter $V/g$, is remarkable. To our knowledge, this is the only known model to show such behavior, at least within numerically accessible system sizes.

### 3.5    Numerical results in the IQS approach

One may look at the dynamics of the state of the DQD on a time scale much beyond $t_{oqs}$ (35), which is of the order of $gt_{oqs} = 5 \sim 8$ for the sizes of $L_B$ we consider ($L_B = 20 \sim 26$); see also Fig. 2 for the value of $gt_{oqs}$. We now do so and analyze the dynamics for times up to $gt = 2000$. Our main result here is Fig. 5 in which we plot the dynamics of one representative observable $\langle \hat{n}_1(t)\hat{n}_2(t)\rangle$. Panel (a) gives the dynamics in the non-interacting, or integrable case $V = 0$. It shows large oscillations which do not decrease in amplitude, even for large times and within the values of $L_B$ considered. In panel (b) we plot the dynamics of the same observable for the interacting non-integrable* DQD model $V = g$. Now the dynamics is drastically different. While the oscillations are sizeable for small times (up to $gt \sim 400$) they are strongly suppressed beyond that time scale, but persist as smaller fluctuations. Those fluctuations decrease with increasing $L_B$. This long time behavior corresponds to small oscillations about a relaxed value. These results suggest that for the non-integrable* model, the dynamical behaviour follows the thermalization statement of the IQS approach, showing fluctuations about a mean value which are exponentially suppressed with increase in lattice size, see Eq.(18). The situation is different for the free (integrable) DQD case, where oscillations persist and thermalization occurs.

Let us now explore the relaxation in the interacting DQD case in more detail, in the light of IQS thermalization, Eqs. (17), (18). We note that in Eq. (18) the time dependence of the integral up to a finite time does not play a role. So, the fluctuations (18) about a time averaged value $\overline{\langle \hat{A}\rangle}$ defined in (17) may be written as

$$\overline{\delta \langle \hat{A}\rangle^2} = \lim_{t\to\infty} \frac{1}{t}\int_0^{t_1} dt' \left|\langle \hat{A}(t')\rangle - \overline{\langle \hat{A}\rangle}\right|^2 + \lim_{t\to\infty} \frac{1}{t-t_1}\int_{t_1}^{t} dt' \left|\langle \hat{A}(t')\rangle - \overline{\langle \hat{A}\rangle}\right|^2, \qquad (42)$$

for any finite $t_1$. The first term on the right hand side converges to zero in the $t \to \infty$ limit, and the second term becomes identical to the infinite time average of the integrand. Therefore, explicitly neglecting the finite time behaviour and defining

$$\overline{\delta \langle \hat{A}\rangle^2}(t_1, t_f) := \frac{1}{t_f - t_1}\int_{t_1}^{t_f} dt' \left|\langle \hat{A}(t')\rangle - \overline{\langle \hat{A}\rangle}\right|^2, \qquad (43)$$

the exponential decay of fluctuations, Eq. (18), can be re-written as

$$\lim_{t_f\to\infty} \overline{\delta \langle \hat{A}\rangle^2}(t_1, t_f) \leq C\, e^{-\alpha L_B}, \quad \forall\, t_1 \geq t^*(L_B). \qquad (44)$$

Here $C$ and $\alpha$ take numerical values specific to the observable $\hat{A}$, while being independent of bath size $L_B$. So, the above expression says that there exists a time $t^*(L_B)$, beyond which the variance of values of $\langle \hat{A}(t)\rangle$ is exponentially small in bath size $L_B$. Our plots in Fig. 5 show that no such time exists for the non-interacting DQD. In contrast, the plots of the interacting DQD show that after some large fluctuations for a short initial time, the variance quickly reduces and remaining oscillations diminish with increasing bath sizes.

We now establish this more clearly for the interacting DQD with $V = g$. For numerical simplicity, we approximate $\overline{\delta \langle \hat{A}\rangle^2}(t_1, t_f)$ by the variance of data points when calculating $\langle \hat{A}(t)\rangle$

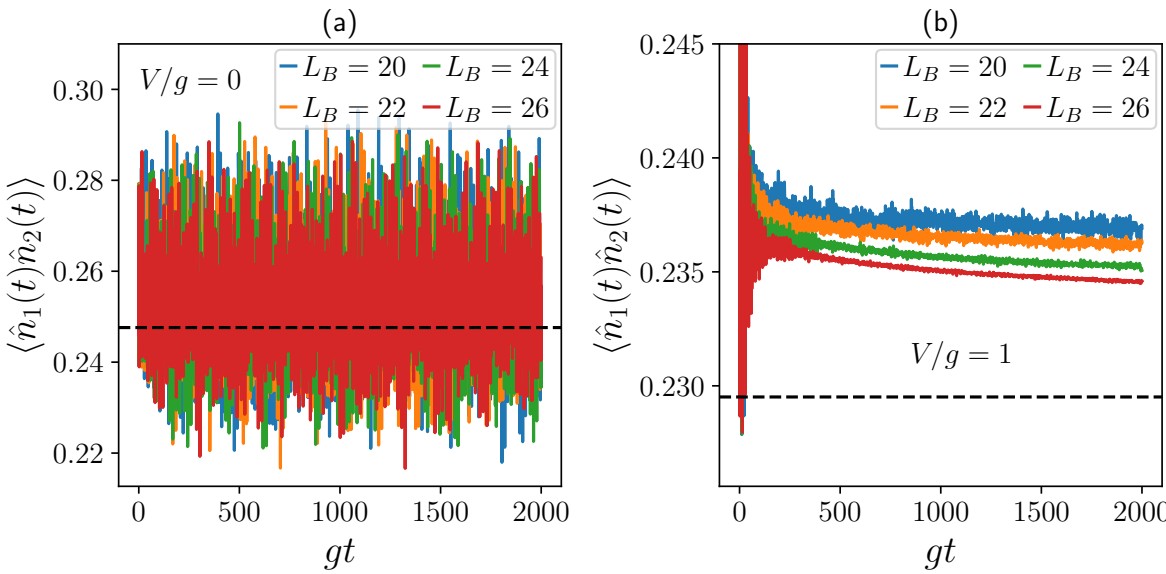

Figure 5: *Thermalization in the IQS regime:* Plots of the dynamics of the representative DQD observable $\langle \hat{n}_1(t)\hat{n}_2(t)\rangle$, starting from a randomly chosen initial state (29) of the DQD. In contrast to Fig. 2, where we considered maximal times up to $gt_{\text{oqs}} \sim 5 - 8$, we now consider times $gt$ up to 2000, including the regime $t \gg t_{\text{oqs}}$. Panel (a) shows the free, integrable, DQD case $V = 0$. Oscillations are found to persist for all times and for all the chosen values of $L_B$. Panel (b) shows the dynamics in the interacting, non-integrable* case $V = g$. The large oscillations disappear after some time $gt \sim 400$ and only small fluctuations about a relaxed value persist. Those small fluctuations diminish rapidly with increasing bath size $L_B$. Other parameters: $\beta g = 0.1$, $\gamma = g$, $g_B = 2g$.

over the time range from $t_1$ to $t_f$. For the interacting DQD, we now plot in Fig. 6(a) the numerical value of the variance $\overline{\delta \langle \hat{n}_1 \hat{n}_2 \rangle^2}(t_1, t_f)$ for fixed $t_f = 2000g^{-1}$, as a function of initial time $t_1$, and for various values of $L_B$ (varying colours). The variance initially decays exponentially with $t_1$. For late times $t_1 > t^*(L_B)$, some time $t^*(L_B)$, the value of $\overline{\delta \langle \hat{A} \rangle^2}(t_1, t_f)$ settles to an approximately constant value, which we denote by $v(L_B)$. The value of $t^*(L_B)$ increases with $L_B$. This, once again, highlights the importance of taking $t \to \infty$ first in Eq. (18).

We also see that the value $v(L_B)$ decreases exponentially with $L_B$, i.e. it obeys the bound $v(L_B) \lesssim C\,e^{-\alpha L_B}$. This is evident from the log-plot shown in Fig. 6(a). To establish this more clearly, we plot the long-time constant values $v(L_B)$ for all four operators of the DQD (see Eq. (25)) as a function of $L_B$ in Fig. 6(b). We see that all these quantities decay exponentially with $L_B$. These observations numerically evidence the exponential decay of the variance with $L_B$ required for IQS thermalization, see Eqs. (44) and (18).

In order to check numerically the validity of Eq. (19), we define for any operator $\hat{A}$,

$$\mathcal{E}\left(\overline{\langle \hat{A} \rangle}\right) = \left| \frac{\overline{\langle \hat{A} \rangle} - A_{\text{MGS}}}{A_{\text{MGS}}} \right|. \tag{45}$$

This measures the deviation of the expectation from its value in the mean force Gibs state, see (20). From (19) we expect $\mathcal{E}\left(\overline{\langle \hat{A} \rangle}\right) = d/L_B + f$ with $d$ specific to the observable $\hat{A}$ while independent of $L_B$, and an offset $f$ which should be negligible. In Fig. 7(a), we now plot $\mathcal{E}\left(\overline{\langle \hat{A} \rangle}\right)$, as a function of $1/L_B$, for all four system operators (25). Here the time average is taken over the

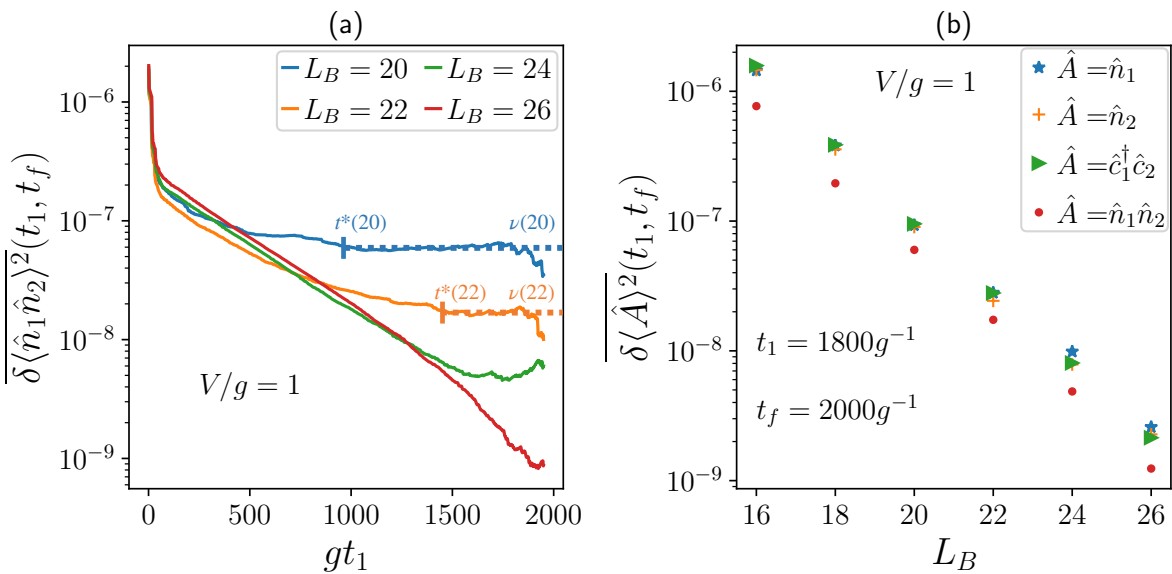

Figure 6: *IQS thermalization of interacting DQD with variable bath size $L_B$*: (a) Log-plot of the variance $\overline{\delta\langle\hat{n}_1\hat{n}_2\rangle^2}(t_1, t_f)$ of data points shown in Fig. 5(b), as a function of initial interval time $t_1$, while keeping the final interval time fixed, $gt_f = 2000$. Here $t_1$ is varied between $0 \leq t_1 \leq 1900g^{-1}$. Different colours indicate results for various bath sizes $L_B$. After a time $t^*(L_B)$, the dynamics reaches an approximately constant value which we call $v(L_B)$ (horizontal dashed lines). The spacing between these saturation values is equal as $L_B$ varies over the values $L_B = 20, 22, 24, 26$, which indicates exponential decay of $v(L_B)$ with $L_B$. (b) Log-plot of the long-time variance $\overline{\delta\langle\hat{A}\rangle^2}(t_1, t_f)$ for each of the four observables $\hat{A}$ of the interacting DQD, as a function of $L_B$. The times $t_1, t_f$ are taken large, so that the variance has reached the approximately constant value (c.f. panel (a)). The quantities $\overline{\delta\langle\hat{A}\rangle^2}(t_1, t_f)$ are numerically evaluated via Eq. (43) with $t_1 = 1800g^{-1}$ and $t_f = 2000g^{-1}$. The plots show that the variance of each observable $\hat{A}$ decays exponentially with $L_B$, in line with Eq. (44). Other parameters: $V = g, \beta g = 0.1, \gamma = g, g_B = 2g$.

same range as in Fig. 6(b). (Note that we have also verified that this plot is almost unaffected by choosing a much smaller value of $t_1$, as long as $t_1 \gg t_{\text{oqs}}$.) We clearly see that $\mathcal{E}\left(\overline{\langle\hat{A}\rangle}\right)$ decays linearly as $1/L_B$ for all choices of $\hat{A}$, with a small offset $f$ (y-intercept).

To see this more generally, we plot in Fig. 7(b) the trace distance between the time averaged density matrix $\overline{\hat{\rho}} := \frac{1}{t_f - t_i}\int_{t_i}^{t_f} dt' \hat{\rho}(t')$ of the DQD, and the state $\hat{\rho}_{\text{MGS}}$, as a function of $1/L_B$. This is done for three different randomly chosen initial conditions of the DQD. We again clearly see a $1/L_B$ decay with a small offset in all cases. Neglecting the small offset, this is completely consistent with Eq. (19). We attribute this small spurious offset to numerical limitations [see Appendix. A].

Combining all of the above, we see that the interacting DQD, satisfies Eqs. (18) and (19), and therefore shows thermalization in the IQS sense for $t \gg t_{\text{oqs}}$. In contrast, the non-interacting DQD does not thermalize in the IQS sense, at least within the range of time and $L_B$ considered. This is consistent with the fact that the interacting DQD shows some non-integrable features in the SFF and the level-spacing distributions, while the non-interacting DQD shows none, as evidenced by Fig. 3. Our numerical results clearly highlight that for thermalization in the IQS sense to hold, we need to first take the long time limit in Eqs. (17) and (18) for a fixed bath size $L_B$, before afterwards considering increased bath sizes. This is the fourth result of the paper.

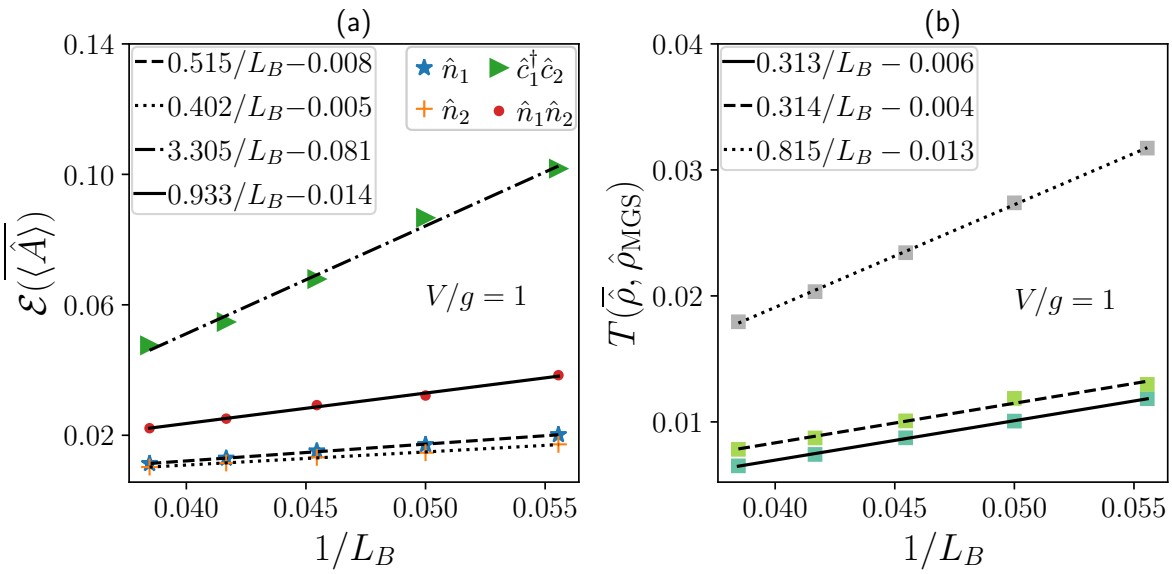

Figure 7: *Decaying deviations from full thermalization in the IQS regime*: (a) Relative deviation $\mathcal{E}\left(\overline{\langle \hat{A} \rangle}\right)$, see Eq. (45), of operator expectation values from their mean force Gibbs state (thermal) values as function of $1/L_B$, for all four DQD operators. Linear fits (dashed, dotted, dash-dotted, solid) show excellent consistency with (19), for all four operators (25). (b) Trace distance $T$ between the time averaged density matrix $\overline{\hat{\rho}}$ and the mean force Gibbs state $\hat{\rho}_{\mathrm{MGS}}$ as a function of $1/L_B$, for three randomly chosen initial states of the DQD (gray, light green, dark green squares). Linear fits (dotted, dashed, solid) show excellent consistency with (19), for all three initial states drawn from (29). Other parameters: $\beta g = 0.1$, $\gamma = g$, $g_B = 2g$. Time averages are taken between $t_1$ and $t_f$ as chosen in Fig. 6(b).

# 4 Summary and outlook

In this work we have carried out a systematic comparison between the notions and conditions of thermalization in OQS and IQS, on the basis of a model consisting of a DQD coupled to a fermionic lead. The notion of thermalization involves taking both the long time and the thermodynamic limits. We show that both the OQS and the IQS approach investigate the convergence to the same state. However, they apply the two limits in a different order: while in the OQS one first performs the thermodynamic limit and then the long-time limit, it is the other way around in the IQS approach. As a consequence, the two approaches give the convergence to the final state at markedly different timescales. This timescale separation in the two thermalization notions, to our knowledge, is a new insight. We believe it stems from the fact that in the IQS approach, the system and the environment are considered a single complex, and relaxation to equilibrium is required at all locations within this complex. In contrast, in the OQS approach, only local variables of the system alone, and not those of the environment, are required to show thermalization, resulting in a faster process.

Here we showed that the OQS regime corresponds to the time scale $t_{\mathrm{oqs}}$ within which the system hardly feels the finite size of the bath. We invoked Lieb-Robinson bounds to show that $t_{\mathrm{oqs}}$ is roughly proportional to bath size $L_B$ for large $L_B$. Using the concept of typicality we have constructed the class of initial states usually considered in the OQS approach, as a special type of a randomly chosen pure state. Starting with such an initial state our numerics show that regardless of the initial state of the DQD, the state of the DQD at time $t_{\mathrm{oqs}}$ converges to

the mean-force Gibbs state $\hat{\rho}_{\mathrm{MGS}}$ as $L_B$ is increased. This corresponds to thermalization in the OQS sense and it holds irrespective of the presence of many-body interactions within the DQD.

We then showed that thermalization in the IQS sense takes place at times $t \gg t_{\mathrm{oqs}}$. In this regime, the DQD is affected by the finiteness of the bath. In this sense, the dynamics of the DQD may no longer be considered as that of an open system and one is lead to view the DQD plus the environment as a single, isolated system. It is known that integrable Hamiltonians do not thermalize in this regime, but according to ETH considerations, non-integrable ones are expected to. The free DQD case corresponds to free fermions, thereby is integrable and is not expected to thermalize in this regime, which we confirm via numerical investigation. However, surprisingly, we found that our interacting DQD model is *partially* non-integrable, i.e, neither integrable nor fully non-integrable, as evidenced by level spacing statistics following a Brody distribution for all interaction strengths. We further found that, despite being *partially* non-integrable IQS thermalization does occur for the interacting DQD case. We thus exhibit a first systematic study of thermalization in a partially non-integrable system, where ETH is not expected to hold. These results evidence that thermalization in the interacting DQD occurs according to both, the OQS and IQS notions, while hinting at the possibility that different aspects of non-integrability may have different influences on thermalization.

Our results open several directions for further research. While here we focused on the behavior of observables defined locally on the reduced system, it will be extremely interesting to extend the analysis to observables defined locally on the bath. Our approach could be applied to other fermionic models than the one investigated here. In this connection, free fermionic models would be specifically of interest. Only a handful of works have insofar considered IQS thermalization in free fermionic systems. In Refs. [89, 90], a three-dimensional disordered free fermionic system was considered. It was shown that the fluctuations about the long-time average decay as a power-law with number of sites, unlike the exponential decay expected for non-integrable systems. Other works on free fermions [91, 92] have defined thermalization as relaxation to an equilibrium state for a finite duration of time that increases with the system size. This definition is more akin to OQS thermalization and differs from the IQS thermalization usually considered in presence of many-body interactions. Clearly, further investigations are required to clarify these aspects in the light of our results. Finally, we introduce the notion of a partially integrable quantum system, and establish thermalization in such a system. It will be of interest to explore the fate of ETH in such a system. Overall, we believe our results call for a more in-depth study of the necessary assumptions for IQS thermalization, as well as, a systematic comparison of OQS/IQS thermalization in a broader class of models.

# 5  Acknowledgements

The authors acknowledge insightful discussions with Jens Eisert and Karen Hovhannisyan. AP acknowledges funding from the Danish National Research Foundation through the Center of Excellence "CCQ" (Grant agreement no.: DNRF156) and Seed Grant from Indian Institute of Technology Hyderabad, Project No.SG/IITH/F331/2023-24/SG-169. AP acknowledges the Grendel supercomputing cluster at Aarhus University, Denmark where much of the calculations were done. AP also thanks Anupam Gupta at Indian Institute of Technology, Hyderabad, for giving access to his workstation where some of the calculations were carried out. JA gratefully acknowledges funding from the Engineering and Physical Sciences Research Council (EPSRC) (Grant No. EP/R045577/1), and from the Deutsche Forschungsgemeinschaft (DFG) under Grants No. 384846402 and No. 513075417, and the Sonderforschungsbereich 1636, project A05 (No. 510943930). MM was supported by a Discovery Grant from the Natural Sciences and Engineering Research Council of Canada (NSERC).

# A  Appendix: Numerical limitations

Although the numerical technique we have used is quite powerful and has allowed us to simulate quite large chain lengths up to considerably long times, there are certain limitations, which can lead to small errors that are hard to reduce. We believe the small spurious offset seen in linear fits of Fig. 7 stem from these errors. Here, we describe the main possible sources of such errors.

First, we used a single typical state to approximate a thermal state. This is done both for the initial condition, and in calculating the expectation values corresponding to $\hat{\rho}_{\mathrm{MGS}}$. While this is a good approximation at large enough system sizes, this is not exact and small sample to sample fluctuations are expected at any finite system size. Deviations between expectation values obtained from a single typical state and those from a thermal state may be reduced by ensemble averaging. But, although they reduce exponentially with system size, for a given system size, they are not expected to decrease as fast with ensemble averaging. Therefore, reducing the deviations at a given system size require averaging over a large number of samples. Since we are performing a long time simulation (up to $t = 2000g^{-1}$), with quite large many-body systems (up to total chain length is 28 sites, size of the largest block of Hamiltonian in Fock space is of size 40116600), using a single typical state is already computationally quite expensive. This makes taking average over a large number of typical states impractical. So, at present, we are unable to make a systematic check of whether the offset can be reduced via ensemble averaging.

Second, in Figs. 5(b), it is apparent that, although there is relaxation, the simulation might not have reached the asymptotic value within the time range considered. It may be that taking time averages over much larger time regimes reduces the offset. This is again hard to check, because going to longer times is also expensive. However, within our time regime, we have seen that taking time average over any time regime with $t_1 \gg t_{\mathrm{oqs}}$ yields plots similar to Figs. 7 with quantitatively similar offsets.

Third, our numerical data is taken in steps of time $gt = 1$. Taking such large time steps is one of the advantages of the Chebyshev polynomial method that we utilize to make such long time simulations tractable within our computational resources. This, however, also means that the average of data points deviates from the actual time average defined in terms of an integral. So, it may be that the spurious offset can be reduced on taking a finer time-grid. But, this again, would increase simulation time greatly, making the simulation impractical.

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
