# Peer review of "On the difference between thermalization in open and isolated quantum systems: a case study"

_SciPost Physics, doi:SciPost Phys. 19, 136 (2025)_

## Round 1 · Referee Report · Jochen Gemmer (Referee 1) · 2024-11-13

Report
The authors address differences between open system vs. closed system perspectives on thermalization. The work is primarily of numerical nature. These numerical investigations are performed for a quantum dot coupled to a lead, the former taking the role of the system the latter that of the bath. It may also be described as a 1-d lattice system of spinless fermions, in which only fermions on the first two sites may interact. If this interaction is tuned to zero, the whole system is integrable. First some theory on open systems and eigenstate thermalization is summarized. Then four results are presented:
-
There is a first temporal regime which is more or less proportional to the length of the bath in which the open system approach is applicable.
-
In this first temporal regime the dynamics of the quantum dot are independent of the size of the bath. If the bath is large enough the mean force Gibbs state is reached. This holds regardless of the system being integrable or nonintegrable.
-
According to the Brody parameter the considered system is never fully chaotic regardless of the interaction strength.
-
In a second temporal regime (basically at all times after the first one) a behavior which the authors associate with isolated system thermalization is found for the interacting case but not for the noninteracting case.
This work is (as the title says) a case study. As such it does not focus on new concepts but brings different concepts together in one example and focuses in the interplay of these concepts. In principle this may warrant publication, however the following points should be satisfactorily addressed prior to possible publication:
-
The title is misleading. The work only focuses on systems that allow for a natural system-bath partition. However, thermalization in isolated quantum systems may also (possibly primarily) apply to systems without any bath. In this case, obviously, open system thermalization simply does not exist/apply Consider e.g. the spreading of heat in a solid. This important difference is not reflected in the title or anywhere else in the paper. Thus the title makes the paper appear more encompassing than it really is.
-
Often the paper emphasizes a contradiction in isolated system equilibration between systems being integrable and nevertheless being equilibrating. This contradiction is over-emphasized. The authors themselves list some counterexamples. However, the discussion of this (at the very end of the paper) is a bit odd: The statement that a form of isolated system equilibration in free fermion systems (as observed in another paper) should be considered as open system equilibration is unjustified, since the free fermion systems do not even have a bath. Attacking the statement "only nonintegrable systems thermalize" is not reasonable, since this is just a flawed oversimplified version of the original eigenstate thermalization as suggested by Srednicki / Deutsch
-
The first result (see 1. above) is not really a result. It is a simple consequence of Lieb-Robinson bounds. It has been already discussed in the same context elsewhere (e.g. papers by Takahiro Sagawa et al.)
-
The third result (see 3. above) may not be too surprising. It is clear that both limits V/g -> 0 and V/g -> \infty are integrable. A system as small as chain length 18 may not reach the fully chaotic regime in between.
-
Specifically interesting is the transition between the two temporal regimes. From the data shown, it appears as if the fluctuations at the (beginning of the) second regime are substantially larger than those at the end of the first regime. To display this clearer it would be helpful to show a plot that focuses on this transition. More concretely, plots that show the transition from the behavior shown in Fig. 2a to the behavior shown in Fig. 5a and the same for Fig. 2d to 5b, would be very valuable.
Recommendation
Ask for major revision
Author: Archak Purkayastha on 2025-07-23 [id 5669]
(in reply to Report 2 by Jochen Gemmer on 2024-11-13)
Warnings issued while processing user-supplied markup:
- Coercing language: reStructuredText
We thank Prof. Joachen Gemmer (JG) for the insightful comments, and for judging the work to be publishable after a revision. We have now revised our manuscript according to the suggestions. Below, we give our point-by-point comments.
JG says
1.The title is misleading. The work only focuses on systems that allow for a natural system-bath partition. However, thermalization in isolated quantum systems may also (possibly primarily) apply to systems without any bath. In this case, obviously, open system thermalization simply does not exist/apply Consider e.g. the spreading of heat in a solid. This important difference is not reflected in the title or anywhere else in the paper. Thus the title makes the paper appear more encompassing than it really is.
Our response
In an extended system without an a priori system-bath partition one is still mostly concerned with observables of a local region only. This gives a natural operational partition into a system (the region observed) and the bath (the remaining degrees of freedom). In our work, we consider observables of the double quantum dot (DQD) | which makes that the system. We believe that in the Referee's example, of heat spreading in a solid, the same picture is true: by measuring locally the temperature at some location, the degrees of freedom of this location constitute the system with the rest being the reservoir, although establishing this will require a separate investigation. Also note that, in our choice of pure typical initial state the DQD and the chain sites are entangled. In this sense there is no partition between the a system and a bath. Rather, dynamics starting from this state can be viewed as a quench from a state perturbed at the first two sites. The crucial point is the following: our pure entangled state is such that the dynamics of the system starting from this state is identical to that starting from a mixed product state between an arbitrary DQD state and thermal state of the chain. This allows the fair comparison between thermalization in isolated and open systems.
It is true that, in our model, the Hamiltonian parameters in the DQD are different from those in the lead. However, our same protocol can be applied to a situation with uniform Hamiltonian parameters, which avoids a separation into system and bath also in this respect. This, indeed, will be an interesting avenue for future research. We have now mentioned this point in the Sec. 4 'Summary and outlook'.
Given these considerations and the fact that our title does not claim all generality by mentioning that we carry out a case study, we would like to keep the title as is.
JG says
2.Often the paper emphasizes a contradiction in isolated system equilibration between systems being integrable and nevertheless being equilibrating. This contradiction is over-emphasized. The authors themselves list some counterexamples. However, the discussion of this (at the very end of the paper) is a bit odd: The statement that a form of isolated system equilibration in free fermion systems (as observed in another paper) should be considered as open system equilibration is unjustified, since the free fermion systems do not even have a bath. Attacking the statement "only nonintegrable systems thermalize" is not reasonable, since this is just a flawed oversimplified version of the original eigenstate thermalization as suggested by Srednicki / Deutsch
Our response
We have now removed the paragraph on the free fermion systems from the 'Summary and Outlook' section. We have also reduced the emphasis on the non-integrable* behavior of the interacting DQD. We have shifted the analysis of integrability breaking in the interacting DQD to the Appendix, and only include a brief discussion of it in the main text. However, since, there has been no previous study of integrability breaking in this model, it remains a new result.
JG says
3.The first result (see 1. above) is not really a result. It is a simple consequence of Lieb-Robinson bounds. It has been already discussed in the same context elsewhere (e.g. papers by Takahiro Sagawa et al.)
Our response
The first result of the previous manuscript said that for \(t \leq t_{\rm oqs}\), the dynamics is same as that of an open quantum system. Indeed, JG is correct, we should not have called this our first result. Some of the authors have themselves used this idea before in developing numerical techniques for non-Markovian open quantum many-body dynamics [see for example, Phys. Rev. B 104, 045417 (2021), J. Chem. Phys. 161, 154105 (2024)]. What we meant was that at \(t_{oqs}\), there is a crossover from open-system-like behavior to isolated-system-like behavior. This crossover in the same dynamics is our first result. We have highlighted this in our revised manuscript.
JG says
4.The third result (see 3. above) may not be too surprising. It is clear that both limits V/g -> 0 and V/g -> infty are integrable. A system as small as chain length 18 may not reach the fully chaotic regime in between.
Our response
While it may not be surprising, it is still a new result. The integrability breaking due to many-body interaction between only the first two sites of a fermionic chain has not been studied before, to our knowledge. As mentioned in the response to a previous comment, we have now removed the numerical exploration of integrability breaking in the interacting DQD from the main text, and shifted it to the Appendix. We have nevertheless done a more thorough analysis of integrability breaking, specifically studying finite-size scaling. It shows that, indeed, at least over a substantial regime of \(V/g\), the system becomes more non-integrable with increasing chain length.
JG says
Specifically interesting is the transition between the two temporal regimes. From the data shown, it appears as if the fluctuations at the (beginning of the) second regime are substantially larger than those at the end of the first regime. To display this clearer it would be helpful to show a plot that focuses on this transition. More concretely, plots that show the transition from the behavior shown in Fig. 2a to the behavior shown in Fig. 5a and the same for Fig. 2d to 5b, would be very valuable.
Our response
We thank JG for this suggestion. Indeed, this is our main result. We have now added a new figure showing this (Fig. 2 in the revised manuscript), and a section discussing the plot (Sec 3.2). We have also highlighted this in various relevant parts of the manuscript.
Attachment:
The revision done by the authors has, from my point of view, increased the quality of the paper substantially . Thus I now recommend publication of the paper as it is.

Author: Archak Purkayastha on 2025-07-23 [id 5668]
(in reply to Report 1 on 2024-11-14)Warnings issued while processing user-supplied markup:
We have now majorly revised the manuscript addressing all concerns of the Referee. We believe the Referee will find the revised manuscript suitable for publication in SciPost Physics. Below we give point-by-point responses to specific comments.
The Referee says
The "first result" of the paper (p.12) determination of time \(t_{\rm oqs}\) seems a trivial consequence of Lieb-Robinson bounds (citation needed). \(L_0\) is set arbitrarily.
Our response
Indeed, the Referee is correct, we should not have called this our first result. It is just a consequence of Lieb-Robinson bounds. Some of the authors have themselves used this idea before in developing numerical techniques for non-Markovian open quantum many-body dynamics [see for example, Phys. Rev. B 104, 045417 (2021), J. Chem. Phys. 161, 154105 (2024)]. What we meant was that at \(t_{oqs}\), there is a crossover from open-system-like behavior to isolated-system-like behavior. This crossover in the same dynamics is our first result. We have highlighted this in our revised manuscript. We have also added relevant citations.
The Referee says
Our response
We are aware of the unfolding procedure. We had plotted properly unfolded level-spacing distributions in our previous version of the manuscript. We have also left out 1% eigenvalues from both ends of the spectrum, which is a rather standard practice, removing effects that come from too low-lying and too highly excited states. We have now explicitly mentioned these. To ensure that we are not seeing any artefacts from the unfolding procedure, we have now also plotted the average ratio of adjacent gaps (see Eq. (47) of the revised manuscript), which does not depend on unfolding. It shows exactly the same behavior as extracted from fitting a Brody distribution.
Although the integrability breaking in the interacting DQD is a new result, in the revised manuscript, following suggestions from the other referee, Prof. Joachen Gemmer (JG), we have removed the highlight from this result. Our main point is to show both the possibility of open system and isolated system thermalization in the same dynamics. In the revised manuscript, all details of integrability breaking in the interacting DQD is moved to the Appendix, and it is only briefly mentioned in the main text.
The Referee says
Our response
As mentioned in our response to the previous comment, we have now reduced the highlight from this result, putting most of the details in the appendix, and only briefly discussing about the integrability breaking in the interacting DQD. We are now cautious to name the relevant sections 'integrability breaking of the interacting DQD'. We have now done the finite size scaling, and indeed, over a large range of interacting strength, it seems that the system becomes more integrable with increasing \(L_B\). So, indeed this could be finite size effect, it is hard to give any more precise conclusion from our data. We have now presented data up to \(L_B+2=20\). This is the largest possible within our numerical resources. This is also quite state-of-the-art in the studies of ETH via spectral properties [see, for example, Phys. Rev. Lett. 125, 070605 (2020)]. Older works going beyond such system sizes with exact diagonalization utilize the translational invariance. In our system, the DQD breaks translational invariance.
Among the various papers shared by the Referee, we find PRB105, 214308 quite useful and relevant, and have also cited it now. The rest of the papers refer to many-body localiztion-delocalization transition / crossover, where both the physics and the motivation are quite different. So we do not find them directly relevant to our present work. Rather, our results are close to the exciting set of recent works suggesting a single impurity in an otherwise integrable model can break integrability, while retaining many features of the parent model. We cited relevant works of the later category even in the previous version of the manuscript [Refs. 78-81 of our present manuscript].
Our point in the previous version of the manuscript was not only that the interacting DQD shows intermediate statistics between integrable and non-integrable. It was that, even though there is intermediate statistics, the system shows thermalization. Contrast, for example, PRB105, 214308, where, it is argued that there will generically be intermediate statistics in a finite size system for small integrability breaking parameter, but, this regime is not expected to show thermalization. Nevertheless, quoting the other Referee (JG), "Attacking the statement 'only nonintegrable systems thermalize' is not reasonable, since this is just a flawed oversimplified version of the original eigenstate thermalization as suggested by Srednicki / Deutsch." Following this comment from JG, we have removed the highlight from this specific aspect of our work.
We stress that, our main point in this manuscript is not the spectral properties of the interacting DQD setting. But rather, it is to compare thermalization in open system and isolated system approaches on the same footing via the dynamics. Also note that our dynamics results are for considerably larger system sizes, which is possible because exact diagonalization can be bypassed.
The Referee says
Our response
The dashed-lines in Fig 4 [Fig 5 of previous version of manuscript] are the corresponding expectation values for the mean-force Gibbs state, i.e, the thermal values. We are sorry for missing mentioning this in the caption in the previous version of the manuscript, and thank the Referee for pointing this out. We have now corrected this.
The reason for the starkly different behavior between free fermionic case and interacting DQD case is due to integrability breaking. However, it is important to note that the many-body interaction is introduced in only one bond. So, this is not the most widely studied case of breaking integrability extensively. There are only a few studies of integrability breaking due to a local impurity [Refs. 78-81 of our present manuscript], and none directly studying thermalization via the dynamics, to our knowledge.
It is also important to appreciate that in the OQS regime, i.e, \(t<t_{\rm oqs}\) (Fig. 3 of revised manuscript), the behavior is qualitatively the same, although quantitatively different, for both free-fermionic and interacting DQD. Both thermalize similarly in that regime. Thus, the stark qualitative difference between thermalization due to integrability breaking in the interacting DQD arise only in the isolated system regime.
From the existing literature on OQS thermalization, it is to be expected that the thermalization of non-interacting and interacting DQD would be qualitatively similar. In contrast, from the existing literature on IQS thermaliaztion, they are expected to be qualitatively starkly different, since integrability is broken in the interacting DQD. This is perfectly in line with what we see. Our main result is showing that the two different notions of thermalization occur in the same dynamics, starting from the same initial state, at different regimes of time and length scales. The regimes are different due to difference in order of taking thermodynamic and long-time limits. This therefore demonstrates the important difference between OQS and IQS thermalization.
In Fig 4(b) [Fig 5(b) of previous version of manuscript], we do not expect the plots to decay with time. We expect the plots to eventually show small fluctuations about a constant value. We expect that this constant value approaches the thermal value, i.e, the dashed value obtained from mean-force Gibbs state, with increase in \(L_B\). These expectations are based on the standard notion of IQS thermalization, and are corroborated in the further analysis, see Fig 5,6 of the revised manuscript and corresponding discussions.
The numerics required for obtaining the dynamics of these reasonably large chain lengths (maximum size \(L_B+2=28\)) up to the very long times we have considered (\(gt=2000\)) is extremely heavy. Yet, such length and time scales are required to demonstrate our main result. So, a full study of the dynamics varying the many-body interaction strength is beyond the scope of the paper. Even without such a study, we believe our main result showing signatures of OQS and IQS thermalization in the same dynamics at different length and time scales is sufficiently interesting.
The Referee says
Our response
We quote the text around Eq.(14) here for easy reference:
" The initial state \(|\psi\rangle\) is supposed to have a small energy spread around the given value \(E =\langle \psi | \hat{H} | \psi \rangle\). This is usually quantified as \(\frac{\langle \psi |\hat{H}^2 | \psi \rangle -\langle \psi| \hat{H} |\psi\rangle^2}{\langle \psi | \hat{H} | \psi \rangle^2}\sim O(L_B^{-1})\) "
This is a standard requirement for equivalence of ensembles in statistical mechanics. It means that the distribution of energy is sufficiently peaked around the mean value for large enough systems. This is expected for any short-ranged system. For reference in the context of equivalence of ensembles in statistical mechanics, please check any standard book of statistical mechanics [For example, "Statistical Mechanics'' by R. K. Pathria, 2nd edition, Sec 3.6. Eq. (4), and following discussions].
For discussion in the context of thermalization following a quench, which is exactly the context we discussed it in the paper, please see Eq.(82) and discussion around it, in the review article: L. D’Alessio, Y. Kafri, A. Polkovnikov and M. Rigol, "From quantum chaos and eigenstate thermalization to statistical mechanics and thermodynamics", Advances in Physics 65(3), 239 (2016). The above is Ref.[50] of our manuscript. We have now cited it before Eq.(14).
The Referee says
Our response
The chemical potential in a microcanonical ensemble is well-defined in standard statistical mechanics. It is \(\left(\frac{\partial S}{\partial N_e}\right)_{E, L}=-\beta \mu\), where \(N_e\) is the number of particles, \(E\) is energy, \(L\) is volume (length for one dimension), \(\beta\) is inverse temperature, \(\mu\) is chemical potential, \(S\) is microcanomical entropy. The above notation means taking the partial derivative of entropy with respect to the number of particles, keeping energy and volume fixed. Given a functional form of microcanonical entropy in terms of energy, volume and number of particles, which can be obtained from the microcanonical ensemble, one can calculate the chemical potential as above.
However, the connection of the above with ETH or IQS thermalization is indeed not well-explored. Thus, since they are not the main points of the present manuscript, we have removed the relevant sentences that had caused the Referee to pause. Instead, in the revised manuscript, we have added a brief clarification on the grandcanonical density matrix in Eq. (33). Then we immediately set the chemical potential to zero, resulting an average over fixed number sectors. This is important for fair comparison between IQS and OQS thermalization because OQS thermalization is usually studied for the case where the initial state of the bath is in a grand canonical ensemble.
The Referee says
What we meant by the quoted sentence is that if the chemical potential is zero, and one wants to obtain expectation values in the grand canonical ensemble, one can get away with considering only the half-filled sector for large enough system sizes. This can be strongly argued for and numerically checked.
However, such a discussion is irrelevant for the present manuscript, since we do take the actual grand canonical ensemble and average over all sectors, i.e. we do not assume a 'sharply peaked' state. The 'superselection principle' here means that there will be no off-diagonal terms connecting various sectors.
In the revised manuscript we have removed the confusing statements, and instead specify the grand canonical state we use for the numerics in Eq. (33).
The Referee suggests
Our response
After the major restructuring of the paper, along with some new results, we believe the Referee will find the concerns addressed adequately.
The Referee says
Our response
We thank the Referee for pointing out the occasional slips. The correct quoted sentence from p.17 of previous version of manuscript should have read: "The situation is different for free (integrable) DQD case, where oscillations persist and no thermalization occurs". We have now made the corrections.
The time \(t^*(L_B)\) is not the Heisenberg time. The Heisenberg time scale is much larger and grows exponentially with system size. It is impossible to reach the Heisenberg time in any practical simulation or experiment of dynamics of large systems, and therefore it is usually irrelevant while discussing thermalization. The Heisenberg time is clearly seen in our plot of the spectral form factor in Fig 7 of the revised manuscript, shown by the dashed vertical line. For \(L_B+2=20\), it is of the order \(W t\sim 10^5\), which corresponds to \(gt\sim 10^4\). This is much larger than the numerically defined time \(t^*(L_B)\) in Eqs. (37), (38) and Fig 5(a). For larger \(L_B\), the Heisenberg time will be even larger.
The Referee says
Our response
We have replaced CPTP by 'completely positive trace preserving' in the text. We have corrected the typo pointed out by the referee. Many of the other short-forms are rather standard, as the Referee also says. As for IQS and OQS thermalization, we feel changing them consistently throughout the manuscript at this stage would be more prone to error. So we would not like to change them.

---

## Round 1 · Referee Report · Anonymous (Referee 2) · 2024-11-14

Strengths
- Well-posed problem, satisfying main criterion 2.
- Generally well written manuscript, however with occasional slips (see below)
Weaknesses
- Unreliable numerical analysis of "nonintegrability"
- Exaggerated claims
- Chaotic and incomplete references, several missing, several redundant
- No system size analysis
Report
The manuscript aims to compare two scenarios of thermalization: an isolated system approach and an open system approach. To this end, a very idealized system is considered in which a fermionic noninteracting lead is coupled to a double quantum dot envisioned as two interacting fermionic sites.
The lead is assumed to form a bath for the double quantum dot. Such a simple model allows for a rather detailed analysis. The authors made several claims about their results:
1. they identify a time scale before which thermalization may take place according to an open system approach;
2. they claim that this time scale is confirmed numerically both for noninteracting and interacting (within the dot) cases.
3. They perform analysis of spectra form factor and spacing distributions claiming that for the interacting dot the system has an intermediate statistics and is a partially non-integrable model''
4. They analyze the dynamics for long times stating the differences betweenopen'' and ``isolated'' approaches linking that to the order of limits taken for time and system size.
Unfortunately, while a comparison between "open" and "isolated" approaches is interesting and has not been studied in such detail before, some of the claims seem either almost trivial or based on insufficient or poorly analyzed evidence.
The "first result" of the paper (p.12) determination of time $t_oqs$ seems a trivial consequence of Lieb-Robinson bounds (citation needed). $L_0$ is set arbitrarily.
Consider the analysis presented in Section 3.4. I do not understand the choice of citations to formulae quoted in pp.(14)-(16) - formulae often originating from either random matrix theory or standard quantum chaos treatments. The reader should be referred to existing good textbooks on quantum chaos (Haake, Stoeckmann) instead of some, rather erratically selected original papers. While these books mainly address single-particle physics, tools such as random matrix theory are universal and were first applied to many-body physics - excited states of nuclei. This physics is almost 70 years old. It is known, in particular, that random matrix theory describes fluctuations around mean values. To get meaningful results one should carefully unfold the spectra (see books above), a procedure which may be tricky (see e.g. PhysRevE.66.036209). There is no mention of the unfolding in Sec.3.4 making the results in Figs. 3-4 doubtful. The mean level spacing appears in (38) but using the argument in (37) x=Wt/2piD where W is the eigenvalues span and D the Hilbert space dimension suggests that the density of states is assumed to be uniform and no unfolding is done. If it is so then the results have to be reevaluated properly. The application of ad-hoc Brody distribution also requires a comment. Again from earlier quantum studies, it is known that it works well for mixed phase space dynamics if a relatively small number of low-lying states are taken into account. Brody fails for highly excited states (see papers by T. Prosen). Why not use Lenz-Haake (PRL 1991) distribution evaluated for 2x2 matrices as Wigner distribution is?
I find the Authors "third" claim "identifying the non-integrable nature of the interacting DQD is the third result of the paper" as insufficiently proven. They themselves quote papers of Modak and collaborators [47,48] who claimed that a "partially nonintegrable" behavior (and e.g. Brody-like statistics) was a finite system size effect and that with increasing size many-body interacting systems become ergodic. Thus a claim of the present authors that their interacting DQD enjoys intermediate statistics and is a first example of such a behavior, should be supported by analysis of different system sizes (different L_B) to show that the observed behavior is not a finite size effect. The work in Section 3.4 is restricted instead to L_B+2=18 sites and no size effects are discussed. Let us note that already Modak et al. considered comparable systems of size 22 and it was ten years ago!. The authors concentrate instead on dependence on V/g which again may be affected by a finite size. Importantly the authors are not aware of or ignore a vast literature on the subject. Here on one side PRB105, 214308 should be consulted both for its contents and for earlier references. Also relevant are studies of ``central spin'' models SciPost Physics 15, 030 (2023) as well as that of impurity induced interactions Phys. Rev. Lett. 126, 030603 ; Phys. Rev. B 105, L220203; Phys. Rev. B 105, 224208; Phys. Rev. B 107, 144201 (2023). Those are in the presence of disorder but for tilted lattices see Phys. Rev. B108 134201. In fact, in the presence of disorder the intermediate statistics appears in MBL transition so a vast number of systems (at finite size) shows the non-integrable nature. What is really the outcome in the thermodynamic limit? - see the recent review arXiv:2403.07111.
Consider now Sec.3.5. While Fig.5 shows a profound difference in the dynamics of noninteracting and interacting systems, no real explanation for that behavior is given. What are the dashed lines in Fig 5(a) and (b)? Is the correlation function in fig.5(b) decaying exponentially or via a power law? What is the dependence on the interaction strength? As it is Fig.5 shows just a profound difference between a noninteracting and interacting cases which is obvious.
Sometimes I simply do not understand what the authors are meaning. In the description of an isolated system thermalization: 1/Why the initial state should fulfil O(L_B^-1) in (14)? What doe it mean usually? May be some reference? 2/I do not understand the statement (same page, just after (16)). " In the microcanonical picture setting the chemical potential to zero corresponds to a half-filling of the entire chain.". The reviewer does not know the notion of the chemical potential in microcanonical ensemble. Similarly the link of the chemical potential (whatever that means) value to the filling. 3/ The next sentence "Convergence to the grand canonical ensemble with zero chemical potential is guaranteed if the state phi is sharply peaked around the half-filled sector." is similarly problematic. If fermions considered are massive then their state psi should correspond to fixed (precisely) particle number and not be "sharply peaked". This is superselection principle.
Requested changes
Improvement of the analysis presented in Section 3. Clear statement of results. Further changes as apparent from the report.
Please read and correct occasional slips. E.g. p.18 "For late times t_1>t*(L_B), some time t*(L_B), the value of ..." t*(L_B) seems nothing else than the Heisenberg time. Please discuss and explain. In p.17 you write "The situation is different for free (integrable) DQD case, where oscillations persist and thermalization occurs." while in p.19 (bottom) "the non-interacting DQD does not thermalize in the IQS sense." As I understand in p.17 it was also IQS approach so these two sentences contradict each other.
Several short forms are used DQD, OQS, IQS,SFF,MGS,ETH,WD. While some of them are well accepted, I would suggest to change OQS thermalization to OST
(remove quantum since anyway everything is quantum in this paper) and similarly IQS to IST. This will help in avoiding statements as (l.6. p7) "analytical proofs that OQS occurs are lacking". I believe the authors meant OQS thermalization as OQS, I believe, exist in Nature without any proof. CPTP is just used once.
Recommendation
Ask for major revision

---

## Round 3 · List of Changes

b. Equation (34) has been added to explicitly state how the thermal expectation values are calculated.
c. Following suggestion from the editor, we have added a few sentences at the end of section 2.2, as well as after equation (36), to explicitly state that in numerical simulation, OQS thermalization effectively occurs even with finite bath sizes.
d. Following suggestion from the editor, we have now added a sentence Sec 3.4.2 (fourth line),
“Note that, all the physics discussed here are for timescales much less than the timescale of Poincare recurrences [84,85], which is expected to scale super-exponentially with $L_B$, and is impossible to reach in any practical numerical or experimental investigation for chain lengths of our interest.”
References [84,85] are newly added citation for recent works on Poincare recurrences in quantum many-body systems.
e. In response to comment from the editor, we have changed the following line:
"To our knowledge, such convergence has not numerically been shown for any system with many-body interactions."
to
"To our knowledge, convergence to the mean force Gibbs state, $\hat{\rho}_\MGS$, has also not been numerically shown for any system with many-body interactions (i.e, non-Gaussian systems)."

---

## Editorial Decision

published